# Photodynamic Therapy and Hyperthermia in Combination Treatment—Neglected Forces in the Fight against Cancer

**DOI:** 10.3390/pharmaceutics13081147

**Published:** 2021-07-27

**Authors:** Aleksandra Bienia, Olga Wiecheć-Cudak, Aleksandra Anna Murzyn, Martyna Krzykawska-Serda

**Affiliations:** Faculty of Biochemistry, Biophysics and Biotechnology, Jagiellonian University, 30-387 Kraków, Poland; aleksandra.bienia@student.uj.edu.pl (A.B.); olga.wiechec@doctoral.uj.edu.pl (O.W.-C.); aleksandra.murzyn@doctoral.uj.edu.pl (A.A.M.)

**Keywords:** photodynamic therapy (PDT), hyperthermia, chemotherapy, radiotherapy, surgical intervention, cancer, combination therapy, hypoxia, drug delivery

## Abstract

Cancer is one of the leading causes of death in humans. Despite the progress in cancer treatment, and an increase in the effectiveness of diagnostic methods, cancer is still highly lethal and very difficult to treat in many cases. Combination therapy, in the context of cancer treatment, seems to be a promising option that may allow minimizing treatment side effects and may have a significant impact on the cure. It may also increase the effectiveness of anti-cancer therapies. Moreover, combination treatment can significantly increase delivery of drugs to cancerous tissues. Photodynamic therapy and hyperthermia seem to be ideal examples that prove the effectiveness of combination therapy. These two kinds of therapy can kill cancer cells through different mechanisms and activate various signaling pathways. Both PDT and hyperthermia play significant roles in the perfusion of a tumor and the network of blood vessels wrapped around it. The main goal of combination therapy is to combine separate mechanisms of action that will make cancer cells more sensitive to a given therapeutic agent. Such an approach in treatment may contribute toward increasing its effectiveness, optimizing the cancer treatment process in the future.

## 1. Introduction: Aims of Anti-Cancer Combination Therapy

Cancer is a disease that is increasingly affecting our society. It is one of the reasons for high mortality across the world. Cancer is a leading cause of death worldwide, and is ranked as first or second in noncommunicable diseases (NCD) in developed countries [1]. In the majority of cases, current cancer treatments are not effective. In 2020, the estimated numbers of new cancer cases amounted to more than 1.8 million, and an estimated 0.6 million cancer-related deaths were predicted [2]. These numbers indicate that 1 of 3 cancer patients die from the disease [2]. Common treatment options include surgical intervention, chemotherapy, and radiotherapy. For some types of cancer, there are new, more popular treatment methods (e.g., immunotherapy for melanoma) [3]. Anti-tumor therapeutic problems include systemic toxicity, pain, reduced immunity, and anemia (it depends on the treatment used). Hence, treatment dosages should be high enough to fight cancer cells, and, at the same time, low enough to prevent serious side effects. The question is: where do cancer cells develop resistance to treatment? The resistance of cancer cells to the administered drugs/radiation and the high toxicity of systemic therapy have contributed to the development of new approaches in cancer treatment. Therefore, attempts have been made to combine various available methods that are used to treat tumors. The main goal of this strategy is to combine different mechanisms of action that lead to sensitizing cells to the next therapeutic factor. Thus, combination therapy is promising for patients, due to the fact that it contributes toward increasing the effectiveness of cancer treatment, and reducing the toxicity for normal cells (as a result of lowering chemotherapeutic doses).

### 1.1. Combination Treatment–Definition, Pros, and Cons

Combination therapy is defined as a form of therapy where a minimum of two standard forms of therapy are applied. The most common combination therapy is, likely, the application of two chemotherapeutics drugs. However, it is the same type of therapy. The most common combination of different types of therapies is the application of chemotherapy and radiation therapy. Combination therapy offers the most effective treatment results. By using this method, greater treatment effectiveness is achieved, and the toxic effects of chemotherapeutic agents is reduced (Figure 1). The results of the applied combination therapy can be additive or synergistic. Their advantages stem from the fact that anti-cancer treatment is targeted at many biological pathways. In addition to cancer treatment, slowing down tumor development, as well as reducing the resistance of tumors to, e.g., chemotherapy or radiation therapy, is also possible. This approach can have much greater results than using monotherapy. It is postulated that cancer cells are often unable to adapt to the simultaneous harmful effects of two therapeutic methods [4]. This requires answering more questions and having deep knowledge about standard therapy and combinatory results. 

In addition to killing cancer cells, targeting different pathways through a combination of treatment techniques can reduce the risk of cancer cells becoming more aggressive and resistant [4]. The undoubted advantage of this strategy is the possibility of achieving such an optimization of treatment elements where, in general, the therapy can be targeted against the particular mechanism. Resistance to the drugs used can be reduced, which was already mentioned before, but it also reduces aggressiveness and metastasis, which contributes to the increase in the effectiveness of fighting cancer. The multi-module approach allows consideration of the heterogeneity of cancerous tumors, which increases the chance of killing all cancer cells. Most importantly, it proves a chance to kill cancer stem cell populations that are known to contribute to drug resistance and cancer recurrence after remission in later years [5,6]. Combination therapies can also reduce side effects by lowering doses of individual monotherapies. Unfortunately, combining different treatments can increase harm to the patient [7]. It is worth emphasizing the further side effects of using photodynamic therapy and hyperthermia. Another important issue is the promotion of metastasis due to increased permeability [8]. However, it is not a simple matter. This is a fairly complex problem. It is important to note that a cancer cell must survive after treatment and be able to pass through the blood vessels within the tumor, and then must survive in a completely different niche, which is not a straightforward matter.

This can make it difficult to identify the agent responsible, and may result in the difficulty to assess which agent’s dose should be reduced. If therapeutic agents work similarly and their side effect profiles are similar, the accumulation of side effects can cause more severe clinical symptoms [4]. The problem (and, at the same time, disadvantage) of using combination therapy is the cost of such therapy. It is widely known that the use of monotherapy is much cheaper than the use of two or even more treatment methods. In addition, there are some questions about the method of maximizing the use of more than one therapy. These questions are related to identification of a “therapeutic window”: what time is best to apply the second stage of treatment in order to maximize benefits? Next, which method should be the leading one, and when should we start each therapy in order to achieve the best results? The issue of using combination therapy is an individual matter and it all depends on the tumor vasculature and the type of cancer, but thanks to the individual approach, and combination of various methods suitable for a given type and stage of cancer, we can achieve surprising results. It is especially important to understand the interaction between two or more anti-cancer agents in a combination regimen to obtain maximum efficacy with the least toxicity. To conclude, the new quality of combination treatment emerges from the selected anti-cancer agents for particular tumors and evolves in the time scale. It requires deep knowledge about treatment mechanisms. On the other hand, there are many combinatory protocols already used in clinics, and they are applied successfully without good understanding of the mechanisms of action.

### 1.2. Action Strategies and the Greatest Successes in the Fight against Cancer

In the context of the use of combination therapies in cancer treatment, there are some very interesting results obtained by the use of chemotherapy, radiotherapy, hyperthermia, photodynamic therapy or surgery, and immunotherapy in various combinations. Currently, a combination treatment of chemotherapy, radiotherapy, and often surgery is used commonly in cancer treatment. With this approach, we can effectively stop cancer development and obtain the best possible results based on current knowledge (but one of three cancer cases still lead to patient death). There is no doubt that combination treatment provides very good results, as opposed to using only one method due to the comprehensiveness of the multi-module treatment. Based on many simulations with the use of computer models, cancer treatment strategies were optimized using a combination of possible treatments [9,10,11]. The crucial elements for optimization are drugs and their doses, as well as the dose of light and radiation, in the context of cancer type and the degree of its aggressiveness. Fractionation of the radiation dose also proved to be more crucial, giving better results [12,13].

In addition, cutting out cancerous tissue, with a margin of healthy tissue, combined with chemotherapy cycles, provides the best results and reduces tumor metastasis. It also allows for targeted treatment and effectiveness increase due to a long-term and selective action. Taking into account the occurrence of cell cycles and cell resistance at various stages of the cell cycle for selected methods, it offers more significant results. Scientific research has also shown that the cells present in the phase that is the most resistant to radiation, i.e., those that are in the S phase (replication phase), are also very sensitive to hyperthermia [14]. Therefore, it was concluded that hyperthermia allows sensitization of cells to X-rays in the further treatment phase [15]. When considering combination therapy, one may come across concepts, such as thermochemotherapy, thermoradiotherapy, and thermochemoradiotherapy. Considering thermoradiotherapy, the general mechanism of action is primarily the possibility of destroying cancer cells that are insensitive to radiation. It also inhibits the process of repairing damage to DNA under the influence of a dose of ionizing radiation. Due to this approach, combination therapy in cancer treatment offers more effective therapeutic results. In turn, analyzing the introduced concept of thermochemotherapy in the treatment of cancer, one may observe an increase in sensitivity and reduction in cancer cell resistance to chemotherapy. With this approach, it is possible to increase the accumulation of chemotherapeutics in the cancer location [15,16]. All of this contributes toward increasing the effectiveness of this complementary method, which, in this context, has been linked to phenomena, such as hyperthermia. Undoubtedly, new classes of chemotherapeutics, as well as various mechanisms of action of drugs, such as the use of drugs that are not dependent on the rate of proliferation, or those that depend on the phase of the cell cycle, and attempts to combine drugs with non-overlapping toxicity, or a use of a combination of drugs that have different mechanisms of action [17], as well as the way of adding drugs, for example using targeted liposomes, significantly improve the effects of treatment and show us the importance of using combination therapy. On the other hand, the use of thermochemoradiotherapy shows that multi-module treatment brings the most beneficial and the best results [18]. Researchers have studied the interaction between hyperthermia and radiation therapy, as well as chemotherapy using chemotherapeutic agents, such as cisplatin. Scientific research has shown that a synergistic effect can be obtained, but only when this three-modular therapy is used on cell lines that are not resistant to cisplatin [19,20]. Considering photodynamic therapy, this combination of therapy with chemotherapy gives quite good results due to direct destruction of cancer cells and strong influence on blood vessels in the tumor surrounding. It was shown that different PDT protocols can significantly damage the vasculature or temporally increase blood flow and tissue oxygenation in the treated area [21]. However, as a result of photodynamic therapy, when not all of the cells are destroyed, the use of chemotherapy in the second wave allows killing the remaining cancer cells that have survived PDT. Additionally, it was shown that PDT can have strong influence on the immune system [22]. With this approach, combining methods can stop the cancer or lead to a complete cure. 

The list of combination therapies (with mixed treatment types) conducted in 2015-2020 was presented in Table 1. The data are presented only for the four most common cancers in the world according to GLOBOCAN 2018 [1] and for pancreatic cancer, which was chosen due to our interest.

## 2. General Information about Photodynamic Therapy and Hyperthermia

### 2.1. PDT

In this article we focus on PDT and hyperthermia treatment as a very interesting example of a combination strategy. That is why this section includes basic information about the selected therapies and their modes of action. Photodynamic therapy (PDT) requires three independent factors: photosensitizer (PS), light at proper wavelength to excite PS, and oxygen. We should note that although a photosensibilized reaction can be possible without any oxygen, a photodynamic reaction involves oxygen by definition. The main advantage of PDT as a cancer treatment is based on its double selectivity: PS is a drug that accumulates better in neoplasm tissues, and light can activate only the PS that is localized in an illuminated area. Ideally, PS and light separately are harmless, and in the presence of oxygen, both of them together can create a deadly weapon—reactive oxygen species (ROS). Photosensitizer can be understand as a pro-drug that requires light to become a drug. When we talk about PDT in the context of cancer treatment, the ideal photosensitizer should accumulate selectively only in cancerous tissue, it should have a minimal toxicity before the light exposure and a high toxicity after using light in a given range of wavelengths. At the same time, it must not cause any phototoxic effects on healthy cells. The photosensitizer should absorb light in the range of about 600 to 1200 nm to perform effectively, in the context of light penetration into the tissues (lower light wavelengths are absorbed by endogenous dyes, longer wavelengths are selectively absorbed by water). There are several subgenerations of photosensitizers [158,159,160,161] categorized by their chemical structures. One representative of the first generation photosensitizer is hematoporphyrin. Porphyrins and porphyrin derivatives are examples of second-generation photosensitizers. Moreover, 5-aminolevulinic acid (ALA), which is used to treat skin cancers, is used quite often in PDT. These compounds consist of four pyrrole molecules that are linked together via methylene bridges. Another example of porphyrin derivatives are chlorines. They are chemical compounds that have modified the double bond in their structure and could absorb light in the infrared spectrum more strongly. Due to the fact that chlorines can be excited by infrared light, they are able to penetrate tissue with light more deeply. These chemical compounds are removed quite quickly from a human body–they need about 24–48 h to be cleared, which indicates less toxicity. Bacteriochlorin ale phthalocyanines are also extremely promising photosensitizers in anti-cancer therapies [162]. The can by excited with red/infrared light; pre-clinical and clinical trials show very promising results. Good examples include TOOKAD against prostate cancer [163,164,165,166] and LUZ11 against head and neck cancers [167], photosensitizers with advanced clinical trials. Light is an inseparable element of a photodynamic therapy. The theory that solar radiation can be used to treat many diseases is very old [168,169]. It was argued that solar radiation could cure diseases, such as albinism and psoriasis. In 1903, Niels Finsen received the Nobel Prize for his work on phototherapy. He used UV radiation to treat skin tuberculosis. Later, photodynamic therapy was used to treat skin cancer. The wavelength should be adjusted to the photosensitizer’s absorption spectrum. Different light sources can be used in PDT, e.g., diode, xenon or halogen lasers [170]. It is worth noting that the longer the wavelength, the deeper its beam penetrates into the tissue. Therefore, for about 700 nm, the depth of light penetration is about 1.5 cm (the light power density decreases in relation to the square of the distance).

The last (but not least) necessary element of PDT is oxygen, specifically molecular oxygen, which is dissolved in the tissues. The presence of oxygen allows the formation of reactive oxygen species, such as singlet oxygen, hydroxyl radical, or superoxide anion radical. The effectiveness of therapy is considered a cytotoxic effect under the influence of oxidation. The combination of these three elements is fundamental when it comes to the principle of PDT. Separately, these three factors are believed to be safe, but their combination has an effect on the cells, it contributes to their destruction, making photodynamic therapy effective, and it is increasingly used to treat various diseases. There are two main mechanisms for the formation of free radicals and reactive oxygen species in PDT. 

The type I mechanism occurs when the oxygen concentration is low and free radical forms are present. This mechanism involves the transfer of an electron or hydrogen atom between the excited photosensitizer and the irradiated tumor tissue. As a result of the photochemical reaction, radicals or anion radicals are being formed. The type II mechanism occurs when the oxygen concentration is close to the physiological concentration. There is a transfer of energy between the photosensitizer, which is in the triplet state, and between the excited form–singlet oxygen.

To sum up the basic information about photodynamic therapy, the method is an alternative way of treating cancer. Among the advantages there are: lower toxicity, high selectivity of the method, and reduction of side effects as compared to chemotherapy or surgery. However, this method also has some disadvantages. Unfortunately, the dependence on an external light source makes PDT mainly suitable for the treatment of tumors on or just under the skin, and on the lining of internal organs [171].

In addition, intravenous administration of PDT may cause systemic toxicity, and due to abnormal angiogenesis, non-specific interaction with blood components and the presence of fibroblasts in cancer foci, only a part of PDT sensitizing substances can be used successfully in this therapy [171,172,173]. Unfortunately, oxygen consumption within the tumor may exacerbate tumor hypoxia, leading to PDT treatment failure [174,175,176]. This method is also quite expensive. The cost of specialized equipment is particularly high. In addition, the selection of an appropriate photosensitizer is very hard. It must not be toxic to the patient and it should give the least possible side effects. It is also worth mentioning that PDT may cause temporary hypersensitivity to sunlight.

### 2.2. Hyperthermia

Hyperthermia is a state in which temperature is elevated above the physiological norm. The hyperthermia therapy involves a controlled increase in body tissue locally or selectively to the needed area. Many techniques can be used to achieve it and each has its own pros and cons. For example, contact heating can be the easiest way to heat superficial tissue; however, it will generate significant temperature gradient, which can be responsible for uneven biological effects. On the other hand, different types of the waves can be used, e.g., ultrasound, radio frequency, or infrared–each is effective based on different tissue parameters (considered pros or cons based on the effects needed). Requirements of a high precision device and advanced dosimeters and treatment planning are for this group of heating techniques. Another hyperthermia induction in the tissue involves a combination of nanomaterials, which can accumulate inside the tissue of choice and be the source of heat, after the activation, e.g., gold nanorods could be activated by the near-infrared light. However, this latter way of heating could be the most precise, but also the hardest to perform because of a wide range of influential factors. It is widely known that elevated temperature informs us about inflammation and diseases, and it is one of the immune system defense mechanisms. The normal human body temperature ranges from 36.2 °C to 37.5 °C [177]. The Table 2 shows the temperature range for specific parts of the human body. Human normothermia is reported to be 36.8 °C [178].

Generally, we can distinguish between general, systemic hyperthermia, and local site-specific hyperthermia. Furthermore, in both types, hyperthermia can be generated by the organism itself (e.g., fewer, inflammation) or can be induced artificially (e.g., irradiation). From the biological perspective, one more classification is essential that is based on the established tissue temperature. Hyperthermia is defined as a temperature ranging from 39 °C and above, where the selective effect of heat on tumors, cancer tissues, promotes killing cancer cells by the influence of high temperature [179]. 

The temperature range from 40 °C to 43 °C is most commonly used in the context of hyperthermia in the treatment of cancer [180,181]. Mild hyperthermia is an increase in temperature in the range from 41 °C to 43 °C [182]. Hyperthermia has been used to treat cancer from ancient times [183]. A lot of research has shown that artificial temperature raising can damage cancer cells. Hyperthermia affects the angiogenesis process and can also damage the structure and reduce the tumor size [16,184]. In this aspect, it is a method of a targeted therapy. However, hyperthermia is used in combination with other methods in order to increase their effectiveness, e.g., by improving drug accumulation [185,186,187]. Very often, hyperthermia is combined with chemotherapy or radiation therapy [188]. 

The main goal of hyperthermia is to sensitize cancer cells to the subsequent treatment method, so it can be understand as a physical adjuvant treatment. It is believed that more sensitive cells should be less resistant to X-rays and chemotherapeutics. For example, cells treated with high temperature may inhibit the repair of drug-induced DNA damage by denaturing the proteins involved in this repair [189]. As mentioned above, hyperthermia can be divided by its location. Local hyperthermia can be carried out using external or internal energy sources, regional hyperthermia through organ perfusion; there is also a whole body hyperthermia [190]. If we consider the principle of local hyperthermia, we locate the tumor area and only heat the area where the tumor is located. To provide the right amount of heat, one may use infrared, ultrasound, or microwave wavelength laser. It is possible to both heat the outside of the skin and introduce appropriate probes with a light source that will generate heat inside the body. In the context of hyperthermia, controlling tissue heating is crucial. To have control over all elements of this therapy, we need to correctly locate a selected heating area, put the heating source in the right place, and constantly monitor the temperature rise. An additional important factor for efficient hyperthermia treatment is the time of the established tissue temperature (plateau) and the time needed to achieve it (ramp). Based on that, the total energy deposition can be calculated. The dose of heat should be safe and should bring the expected results without, e.g., painful burns. It should be highlighted that this method can have various side effects. The most common side effects are burns, swelling, and bleeding. Therefore, hyperthermia needs to be applied carefully and should act very selectively, and the heating area should be estimated very well. It should be mentioned that different kinds of tissue will heat in a different way. Fat will heat and accumulate the temperature differently than a well perfused muscle. In this context, tumors are very special objects to heat because lack of tissue homeostasis results in pathological mechanisms of tissue cooling, e.g., uncontrolled perfusion that slows down heat dispersion, specific vessel structures that affect the possibilities of vein and artery heat enlargement. Hyperthermia may prove to be a successful adjunctive therapy for drug transport in which nanoparticles are used. An example of nanoparticles that can aid drug transport is fullerene, which can target the delivery of drugs to the tumor area [191].

Although hyperthermia is usually used as a supportive method, it can contribute to cell death directly by causing damage to the protein–lipid cell membrane as well as cause denaturation of intracellular proteins [180]. Hyperthermia also leads to dismantling, denaturation, and reorganization of cytoskeleton proteins [192]. As mentioned, hyperthermia can also lead to degradation of proteins responsible for repairing DNA damage caused by chemotherapeutics and, thus, inhibit the repair of DNA double-strand breaks in the process of homologous recombination [193,194]. High-temperature cell death can occur through both apoptosis and necrosis. The putative biological–molecular mechanism of hyperthermia is based, among others, on the expression of heat shock proteins (HSPs), induction and regulation of apoptosis, and signal transduction and modulation of drug resistance in cells. Hyperthermia can lead to cell death directly. Induction of cell death may be dependent on the p53 suppressor protein. It is also believed that apoptosis is mainly caused by activation of procaspase 2, which entails the entire cascade of Bax- or Bak-type proteins [181].

Hyperthermia also affects the immune system. In the case of local tumor hyperthermia, an influx of NK cells and macrophages to the tumor location can be observed where local heating is applied. Moreover, it is said that hyperthermia activates the immune system and generates an immune response. HT helps to increase the level of TNF-alpha. A fairly quick response from the immune system with the secretion of cytokines, such as TNF-alpha and IL-1-beta, contributes to the activation of the immune system defense mechanisms [195]. Hyperthermia is currently being modified to achieve the best results with better safety. It is proposed to use nanogold to monitor the course of hyperthermia. In addition, this technique is being optimized to heat the tumor location more efficiently, and to obtain a correspondingly long therapeutic window to introduce the drug into the tumor area more effectively, or irradiate the tumor area with X-rays. In general, nanogold particles are supposed to absorb infrared light to heat the area in which the nanogold has been located [196]. To achieve better therapeutic results, hyperthermia is being modified and new possibilities of treatment are being created.

## 3. Why Are Hyperthermia and PDT Used in Treatment? Why Are They a Good Option for Combination Treatment?

Hyperthermia and photodynamic therapy are both used in treatment since, when performed well, they provide relatively small side effects as compared to chemotherapy or radiotherapy. Undoubtedly, the dose administered is critical here, however, PDT and hyperthermia compared to other conventional methods are less toxic, less harmful, and more secure [197]. The reasons why PDT and hyperthermia should be considered as a good option for combination treatment are the following:

### 3.1. Selectivity

When it comes to the selectivity of these therapeutic methods, the photosensitizer in PDT and nanoparticles used for hyperthermia, e.g., gold nanorods, are of significant importance. These compounds locate more intensively in tumors than in the surrounding tissues. The light needed for PDT and hyperthermia can be used in selected regions, and the production of heat or reactive oxygen species will be intense in the target region where the appropriate wavelength of light is applied. Sensitivity to these factors may depend on the oxygen concentration in the environment, pH, or the structure of the photosensitizer and other nanoparticles. The photosensitizer accumulates at a specific location of the lesion, and then it is activated by an appropriate wavelength of light and, thus, an excited form is performed. The resulting energy can be used to create reactive oxygen species and then a photodynamic reaction (ROS cascade) takes place.

Very often, in the context of PDT results, the concept of ablation of lesions only within a pathological tissue, which is also a result of the selective operation of this method, is taken into consideration. In addition, it also depends on the specific photosensitizer that will be used during photodynamic therapy, and how selectively it will be excited using the selected light source.

Nanoparticles needed to induce some form of hyperthermia also have the ability of selective accumulation in the cancerous tissues. 

### 3.2. Precision

The light source used in these therapies plays an important role in achieving the precision of PDT and hyperthermia. Lasers with high beam collimation can be used for hyperthermia and PDT, and they can illuminate cancerous lesions very precisely. Red light is being used most often, and it can deeply penetrate the tissue. However, precision fiber optics application is opening new possibilities for blue and green light therapies (with minimalized invasiveness). This allows using PDT and hyperthermia to treat both surface lesions and tumors inside the body. According to various studies, good therapeutic effects can also be obtained by using light with a much shorter wavelength. Especially in the context of treating surface changes, blue light with a wavelength of about 400 nm allows penetration of up to 1 mm, which is quite a sufficient effect. As a result, we work on the skin optimally and do not penetrate deeper skin layers. The process of optimizing PDT and hyperthermia is also important for the precision of the entire treatment. Adjusting the light source, light length, and power, depending on the patient’s needs is extremely important. In terms of combining hyperthermia and photodynamic therapy, it is important to choose the appropriate temperature range in which we heat the tissue locally. An exemplary mechanism is the combination of PDT and hyperthermia using porphyrins, which are an interesting choice as photosensitizers [198]. They are used to treat bladder cancer, and they can effectively produce mitochondrial ROS [199].

The photosensitizers used are toxic when excited by light; therefore, patients are sometimes recommended not to expose to light, e.g., for one day for ALA. They may be slightly photosensitizing even without the use of light [200]. The ideal option in PDT are nano-photosensitizers that can precisely reach selected places [201]. In general, PS is safe and non-toxic when not exposed to light. The use of optical fibers and probes that are important in PDT is also of a great significance. They allow access to areas that are hard to reach, e.g., inside the body. This strategy is very successful for the treatment of prostate cancer [202]. It should be mentioned that a well-performed PDT allows reducing tumors without undue damage to properly functioning tissues. A significant increase in tumor response was also observed. It was found that the anti-tumor effect depends on the type and dose of the heat applied, type and dose of the light with a specific wavelength used, as well as the photosensitizer (e.g., mechanism of the photodynamic reaction, cellular localization, biodistribution, and pharmacokinetics), and the sequence of action (time gap between hyperthermia and PDT, time duration between PS administration, and illumination). When it comes to optimizing photodynamic therapy, the dose of the photosensitizer and laser power are crucial.

The phenomenon of hyperthermia allows the cells to be sensitized in a very precise way in order to minimize any possible side effects. Hyperthermia involves providing a proper portion of energy that will heat the local tissues or the tumor. The temperature is usually increased during therapy, from 40 °C to about a maximum of 45 °C, which allows safe heating with little risk of damage to healthy, properly functioning, non-pathogenic tissues [203]. What distinguishes healthy tissues from cancer cells is the fact that healthy tissues are excellent at temperature regulation and adequate heat exchange. By contrast, cancer cells and tissues within the tumor are not able to efficiently dissipate heat due to poor blood supply; therefore, the heat is retained and accumulated, which damages the structure. According to research, cells in the S phase, i.e., in the replication process, are more susceptible to hyperthermia [204,205]. In the context of the hyperthermia mechanism, this method can be divided into external and internal. In the treatment of tumors on the skin, applicators that generate heat within the lesion are used, and thanks to this, the heat is directed into the cancerous area, and the temperature is increased only within the applicator. However, when it comes to the internal heating of, for example, tumors located within important organs, probes are used to bring heat safely and to only heat the specific, designated area inside the body. Due to the fact that heating control in surrounding tissues can be problematic, the nanoparticles are used to increase heating precision by selective nanoparticle accumulation. The next strategy is to use different tissue proprieties, e.g., conductivity, to heat specific tissue type, e.g., by radiofrequency irradiation. 

### 3.3. Broad Effects on Tumor Cells

It is worth emphasizing that hyperthermia, just like photodynamic therapy, has a multifactorial effect on cells. Both PDT and hyperthermia as forms of combination cancer treatment seem to give the best results due to synergistic effects that arise when such two-module treatments are applied. The general scheme of hyperthermia and photodynamic therapy is based on the fact that by increasing the temperature and heating the area where cancer cells are present, which occurs when hyperthermia is used, we increase the amount of mitROS at the same time. This shows that treatment with hyperthermia enhanced the cancer cell-specific PDT activity by increasing the level of mitROS, which led to a reduction in the expression of efflux proteins ABCG2 [206], thereby leading to the accumulation of compounds inside the cell due to the reduced levels of efflux transporters. Another example is the interaction between the hyperthermia used during photodynamic therapy treatment and the presence of the hematoporphyrin derivative. Subsequently, we can examine the following interactions: hyperthermia inhibits the repair of damage that arose as a result of photodynamic therapy, i.e., we can say that there is a cumulative effect aimed at maintaining the damage caused by PDT to achieve better therapeutic results. HpD can accumulate in the tumor, and then it will be excited with laser. During photodynamic therapy and when HpD accumulation occurs at the time of treatment, when we use additional hyperthermia as another treatment method, a synergistic effect that allows increasing the effectiveness of the treatment can be observed.

The molecular mechanism of cell death in PDT depends on the PS accumulated dose [207], localization of the PS inside the cells or tumor compartments [208,209], PS mechanism of actions [162], oxygen availability [210], light dosage [211], type and places of post-ROS damages [212], cells, and tissue repair potential [213]. Such a mechanism can directly induce cancer cell death. As a result of PDT and hyperthermia, mitochondrial reactive oxygen species (mitROS) are produced, which increase the level of HpD (hematoporphyrin derivative) accumulation in the tumor area. It has been shown that an increase in HpD accumulation stimulates and increases PDT activity. MitROS influences the regulation of the HCP-1 expression (heme carrier protein 1). The aforementioned HpD is a derivative of hematoporphyrin, and in turn, HCP-1 is a heme 1 carrier protein. This protein enables the transport of porphyrins. The use of the two-module therapy, where hyperthermia is applied first, will increase the production of ROS [214], which is important in terms of cancer cell mortality. However, the results indicate that 20-min PDT, where the laser power is 200 mW/cm^−2^, and the hyperthermia ranges from 41 °C to 42 °C for about 10 min has some impact on the damage of cancer cells and, consequently, causes their death [215]. One may notice its consequences when it comes to the expression of HCP-1 due to the fact that this expression increases, and this allows for more effective transport of porphyrins and their accumulation in cancer cells, which in turn increases the effectiveness of PDT [206]. Furthermore, it has been shown that increasing mitROS not only leads to an increase in HCP-1 expression, but also downregulates the expression of ABCG2 proteins [206]. Overexpression of, among others, these transporters by cancer cells has been identified as a key factor in the development of resistance to chemotherapeutic agents. In addition, ABCG2 protein (BCRP) has been shown to excrete porphyrins to maintain intracellular porphyrin homeostasis [216]. Tumors can be destroyed using PDT in the following ways: firstly, apoptosis, but also necrosis may occur. Another interesting mechanism of cell death in the context of PDT is the role of autophagy [217,218]. Photosensitizers can be given intravenously, intratumorally, or as an ointment to the skin, but they are effectively absorbed and they penetrate quite well. In the context of necrotic areas inside the tumor, researchers have found evidence of PS accumulation in a dose-dependent manner. A high dose during photodynamic therapy, as well as a fairly high concentration of photosensitizer, generally induces cell death in the necrosis pathway. In contrast, low doses of both photosensitizer and light promote cell death through apoptosis [219].

Hyperthermia at the cellular level allows for the induction of cellular stress. In addition, heat can affect cells by damaging them, causing cell death, or elevating their temperature, it can also activate their defense mechanisms [205]. This phenomenon has a significant impact on cells and protein induction. As a result of hyperthermia, heat shock proteins (HSPs) are being induced [220]. It is worth noting that cell membranes are extremely exposed to heat and can be damaged [221]. Hyperthermia primarily reduces the cells repair capacity and, as a consequence, the therapeutic effect increases. 

### 3.4. Increased Blood Flow in the Tumor 

PDT and hyperthermia also affect blood vessels and blood flow. 

It is also postulated that some PDT protocols may induce increased perfusion in tumor tissue due to cell death (this leads to a decrease in the number of cells and decrease in intratumoral pressure) [222] and also leads to stimulation of angiogenesis, as well as angiogenic signaling [223] (e.g., oxygen consumption as a result of a photodynamic reaction may induce an anti-hypoxic mechanism).

As for hyperthermia, the generated heat artificially increases blood flow through the vessels and, thus, affects the drugs used in subsequent treatment. For example, when used in chemotherapy cycles, it is easier for chemotherapy drugs to enter neoplastic cells [191], and the much higher temperature and the increased blood flow sensitize neoplastic cells to chemotherapy [185]. 

### 3.5. Vessels Pruning and Hypoxia

Some damage to the vessels inside the tumor can lead to damage the tumor itself. In addition, it is important to consider the possibility of changes in the mechanism of action in combination treatment. It must be noted that conducting such a multi-module treatment can give different results and affect the damage of cancer cells or the microenvironment in a different way. To avoid unpredictable effects, the mechanism of the combination treatment should be investigated in vitro and in vivo. It should be mentioned that both PDT and hyperthermia can have dose- and time-dependent results. In the context of cooperation with any form of chemotherapy, hyperthermia can enhance delivery of a drug to an area in which the flow through the blood vessels is more effective. This effect can be used to deliver the photosensitizer to the poorly perfused tumor areas. In the outcome, PDT can be more effective [224]. It is believed that a poorly perfused area can be hypoxic too, and oxygen deficiency can limit the photodynamic reaction. However, changes in blood flow can affect tissue oxygenation as well [225].

In specific cases and conditions, blood vessels may rupture under the influence of photodynamic therapy, especially when the tumor is highly vascularized. This can disrupt the supply of nutrients and oxygen to the tumor. For example, PDT can enhance tissue necrosis or induce tissue reoxygenation [21]. Attention should also be paid to the influence on tissue perfusion. Treating patients with PDT hypoxic tissues is still problematic (due to the fact that the molecular oxygen has to be present to perform photodynamic reaction). However, it is postulated that a new generation of photosensitizers and modifications can help solve this problem [226]. It was observed that PDT treatment can induce better oxygenation of cancer at a specific time after treatment and create a “therapeutic window”. In this particular time after PDT, some additional therapy, e.g., radiotherapy can be much more effective thanks to the oxygen enhancement effect.

The tumor microenvironment is also critical to hyperthermia. During hyperthermia, i.e., when the temperature in the tumor area increases, blood flow through the vessels may significantly increase, at the same time increasing blood vessel perfusion. This causes an increase in the concentration of nutrients and oxygen; therefore, the area of hypoxia may be eliminated at least for a while. In addition, the tumor hypoxia occurs quite often (low partial pressure of oxygen, generally <10 mmHg). The hypoxic area can become more oxygenated due to mild/moderate hyperthermia treatment: heating to several degrees increases the blood flow to the tumor and, thus, increases the oxygen level [227]. Such a pre-treatment can make cells less resistant to radiation therapy. Hyperthermia makes cells more sensitive to radiation [15]. Hypoxia is also crucial in the context of hyperthermia. Hyperthermia can make cells more sensitized and, hence, cause the area of hypoxia to be lifted; this facilitates cell destruction due to hyperthermia. 

### 3.6. Inflammatory Reaction against Cancer

Additionally, PDT can induce local inflammation. In this case, it stimulates various pro-inflammatory substances [228]. PDT causes neutrophilia to migrate to a tumor treated with photodynamic therapy, which is associated with the expression of chemokines as well as interleukins [229]. 

Under the influence of photodynamic therapy, when light is applied and reactive oxygen species are formed, blood vessels dilate, which stimulates the secretion of cytokines and proteins, as well as various growth factors and mediators associated with triggering inflammatory and immune responses. This will stimulate neutrophils and macrophages to migrate to the area where PDT is applied. As a result, according to the research, cancer cells are to be destroyed faster. T-CD4 and T-CD8 lymphocytes, which are known to be cytotoxic, are also activated here. 

Thanks to this, the mechanism of apoptosis, i.e., programmed cell death, is activated. PDT, just like hyperthermia, can alert the immune system as well. Increased temperature may lead to increased HSP expression, which also contributes to the activation of the immune system [230]. Significantly increasing the temperature using hyperthermia, it is accompanied by a state of necrosis [231]. As a result of the increased temperature, HSP 70 is activated, which results in the activation of the immune system [232]. In the case of mild hyperthermia, apoptosis is more common [233].

### 3.7. Induction of Long-Term Immune Response against Tumor

The long-term immunological aspect cannot be forgotten either. During cancer treatment, it is very important to remember the immune response [234]. Increasing the temperature stimulates the immune system and has a positive effect on defense mechanisms, which helps in the fight against the disease [235]. It is worth emphasizing that, on the immunological level, hyperthermia allows the activation of immune cells and expression, and influences MHC, both class II and class I [235]. Among the mechanisms present under the influence of the elevated temperature, the role of hyperthermia in regulating expression by increasing MHC class II, CD80, CD86, and CD40 in DCs is worth mentioning [236]. The fact that dendritic cells are involved in this process is also of great significance. NF-κB is activated on macrophages and dendritic cells, induced by LPS. Additionally, the production of cytokines is activated [237]. MHC class I expression on the surface of neoplastic cells is increased [238]. Moreover, the expression of TLR4 is increased as well [235].

### 3.8. Targeting Possibility

The photosensitizers used in PDT and the nanoparticles used in hyperthermia can be targeted to a specific biological mechanism, while at the same time being compatible with the subsequent treatment modality. In the context of combination therapy, chemotherapeutic substances may accumulate more intensively in the neoplastic tissue due to a “therapeutic window” [21]. This method is selective and the issue of combining photosensitizers with nanostructures is increasingly discussed [239,240]. An alternative “therapeutic window” mechanism involves creating perfusion/oxygen enhancement in neoplastic tissue. The emergence of various types of nanostructures in cancer therapies, including PDT and hyperthermia, increases the potential for designing effective drug delivery methods to target sites. Such particles may be lipid and polymer nanostructures, nanometal particles, which have the property of attaching targeted ligands, therapeutic agents to their surface, or they may be carriers for therapeutic agents. The use of targeted nanoparticles has the advantage of reduced systemic toxicity [241]. Drugs can be designed to target specific biological structures, such as VEGF receptors [235]. 

One of the challenge of PDT is to optimize treatment to balance tissue ablation and perfusion enhancement effects. One way to achieve this is to work with the time period between photosensitizer injection (e.g., systemic, intratumoral) and illumination. Therefore, the photosensitizer and light dose can induce different types of effects, e.g., vascular targeted photodynamic therapy (VTP) and cellular targeted photodynamic therapy (CTP) [21]. Combined phototherapy with other therapies, e.g., chemotherapy, may bring synergistic therapeutic effects. A frequent target of targeted neoplastic therapies are receptors located on the membranes of neoplastic cells, which are responsible for cell signaling, including various metabolic pathways. They can be inhibited or activated by nanoparticles. However, this is not so simple because of the tumor heterogeneity that makes it difficult to successfully target cancer. Neoplastic cells within the tumor differ from one another, which may mean that the response to a given therapy may differ from cell to cell. This is due to differences in the structure of surface receptors, the expression of receptors on the surface of the cell membrane, the number of receptors, and the fact that different types of cells have completely different sets of receptors. With nanomedicine, it is possible to circumvent this problem by amplifying the tumor target. This can be done in four ways: by increasing the number of existing target molecules, e.g., surface receptors, by targeting the therapy in a double-targeting method, i.e., simultaneous action on two tumor-specific factors, by introducing artificial receptors that are new targets for therapy, and by modifying proteins and peptide building receptors in order to increase their affinity to ligands, which can be used in targeted therapies. Thanks to this type of solutions, we can reduce metastasis and drug resistance of cancer cells [241]. 

### 3.9. Nanomedicine

Currently, nanomedicine is also used in the context of cancer treatment. It is a very promising field of science. Nanomedicine allows increase of the effectiveness and improves the safety of the standard treatment and diagnostics [242]. Both PDT and hyperthermia use certain aspects of nanomedicine that allow, inter alia, effective treatment, including more effective drug delivery [243]. Research related to the creation of nanoconjugates by combining many chemical compounds, including drugs, with nanoparticles, which are transported and delivered much better to the preferred area of the tumor, is intensively conducted. Additionally, some photosensitizers and nanoparticles used for hyperthermia can be designed to aim at specific biological targets, e.g., VEGF receptors [244]. This brings the opportunity of targeting specific biological mechanisms that can be comprehensive with the next treatment method. Another important biological effect is increasing receptor expression for active targeting, including nanomedicine [245]. 

The use of nanomedicine in PDT translates into a significant improvement in biodistribution of the photosensitizers used and allows very selectively to reach the tumor area, which directly increases the effectiveness of treatment [200]. In the case of PDT, fullerenes can be used, which are commonly known and can act as photosensitizers. Fullerenes as carbon nanomaterials are a great option when it comes to using these materials as a drug carrier [246]. Additionally, liposomes are used in PDT to transport photosensitizers. Liposomes help to reduce the amount of side effects and reach the tumor area more effectively. This is a common nanoscale modification of drug delivery that increases efficacy and benefits drug pharmacokinetics [247]. Moreover, PDT also uses nanotechnology where photosensitizers are modified by adding quantum dots to the system, which also seems to be a great solution [248]. It connects to a photosensitizer, and the task of quantum dots is to target and deliver photosensitizers directly to cells, providing additional enhancement in drug transport to the desired therapeutic area [249].

Hyperthermia is also of great importance in nanomedicine. Therapeutic protocols are constantly being improved and treatments using this technique are modified. The main focus is on the use of nanoparticles to influence local or systemic heat generation. For this purpose, gold nanoparticles can be used, which are characterized by the ability to absorb light in the near-infrared range, which allows the tumor to be heated. Moreover, it is also postulated that nanogold can increase the doses of X-ray radiation in the tumor area [250]. Another very interesting nanoparticle can be silver, whose task is also to generate an appropriate amount of heat, which allows the drugs to penetrate deeper into cells or tissues [251]. On the other hand, silver nanoparticles can also be used in specialized wound dressings resulting from hyperthermia [251]. Additionally, ferromagnetics are used in the case of magnetic hyperthermia. Ni-Cu is an example of a molecule that can be used for this purpose. Low concentrations are not toxic, in high concentrations, a toxic effect on neoplastic cells is observed [252]. Another interesting aspect of nanomedicine and nanotechnology is the use of nano thermometers that can determine the temperatures of solutions and biological systems, even at the cellular level [253]. The general principle of operation of such a thermometer is based on the use of nanoparticles that are thermosensitive to fluorescence and capture the change in fluorescence intensity [253].

All this means is that PDT and hyperthermia contribute toward increasing the effectiveness of cancer treatment. These are complementary methods, since their use reduces the side effects of drug delivery in the case of chemotherapy, and reduces cell resistance to radiation, but also because these methods are definitely less harmful by themselves [238]. We claim that these therapies seem to be a good solution for combination therapy and should be explored by clinicians more deeply. Based on scientific evidence, these methods undoubtedly increase the effectiveness and efficiency of therapy. In summary, the phenomenon of PDT and hyperthermia treatment is similar—both therapies can alert cancer on a cellular level and induce cell death. These therapies can also impact tissue perfusion and influence tissue oxygenation (hypoxia) through reducing or increasing the nutrients. They can also induce a local and systemic immune reaction.

## 4. What Other Methods Are PDT and Hyperthermia Combined with?

PDT and hyperthermia as an anti-cancer treatments are most often associated with radiotherapy, chemotherapy, immunotherapy, and surgery (Table 3). These methods can be applied in various combinations. The combination of only two methods and a multi-module treatment can be equally effective. It all depends on the type of cancer, its aggressiveness, metastasis, location, and the therapeutic effect we want to achieve. In order to select the appropriate and optimal treatment methods to achieve the best possible results, all of these factors should be considered [254,255].

The treatment time window should be taken into very careful consideration. In general, a “treatment window” induced by a therapy occurs and lasts for a specific time after the treatment [256]. A treatment combining hyperthermia, chemotherapy, and/or radiation therapy is quite often used in clinics. In this connection, hyperthermia primarily plays the role of reoxygenation of cancerous tumors. Thus, tumor reoxygenation may enhance a positive response to radiotherapy [257], as well as sensitize cells to chemotherapy. It was also shown that, in pancreatic cancer, the combination of gemcitabine with cisplatin and hyperthermia gave better results than monotherapy [258]. In contrast, a triple modal treatment was used in a clinical trial of an advanced cervical cancer. It was shown that 74% of patients survived without any recurrence symptoms, and an additional 5-year overall survival rate amounted to 66.1% [259,260]

PDT, on the other hand, can be used in conjunction with surgery as a neoadjuvant, adjuvant, or repetitive adjuvant treatment, and it is also successfully combined with radiation and chemotherapy. For example, apoptosis-modulating factors, such as rapamycin, Bcl-2 antagonists, and others have been shown to increase PDT-mediated cancer cell death. The combination of PDT and radiation therapy leads to the strengthening of anti-cancer effects. For example, the combination of photodynamic therapy and high-dose brachytherapy in patients with symptomatic lung cancer was tolerated well and gave satisfactory results [261].

**Table 3 pharmaceutics-13-01147-t003:** Combined clinical therapies, including hyperthermia and photodynamic therapy.

Hyperthermia in Combinatory Anticancer Treatment								
Schema of Combinatory Treatment	Cancer	Stage of Trial	Country	Study Start Date–Study Completion Date	Enrollment	Results of Trials	ClinicalTrial. Gov ID	Ref.
**H (FRWBH) + R + CH**	Head and neck cancer	1 phase	Germany	2018–2020	10	The median LRC and OS of all included patients * was 10 and 9 months, respectively. * Due to COVID-19 only 5 patients received all cycles of Fever-range whole body hyperthermia (FRWBH).	NCT03547388 [262]	[263,264]
**H + CH + S** **(HIPEC + CRS)**	Gastric cancer	1 phase	United States	2016–2018	4	—	NCT02672865 [265]	—
**H + CH**	Bladder cancer	1 phase (early)	United States	2008–2011	15	With a median follow-up of 3.18 years, 67% experienced another bladder cancer recurrence (none were muscle invasive) and 13% experienced an upper tract recurrence.	NCT00734994 [266]	[267]
**H + CH + S** **(IPHC + CRS)**	Appendix cancer colorectal cancer	1 phase	United States	2007–2007	16	—	NCT00458809 [268]	[269]
**H + S + CH** **(HIPEC)**	Peritoneal cavity cancer	1 phase	United States	2007–2011	17	—	NCT00625092 [270]	—
**H + CH + LS**	Lung cancer	1 phase	United States	1999–?	—	—	NCT00020007 [271]	—
**H + TT**	Liver tumor	1 phase	United Kingdom	2014–2017	10	—	NCT02181075 [272]	[273,274,275]
**S + HIPEC + IT**	Ovarian adenocarcinoma fallopian tube adenocarcinoma, primary peritoneal carcinoma	1 phase	France	2011–2015	30	With a median follow-up of 29.3 months since the diagnosis and 23 months after CCRS + HIPEC. Median DFS from CCRS + HIPEC was 16.7 months and after CCRS + HIPEC, 2-year DFS and OS were 27% and 71%, respectively. The median PFS was 16.7 months after surgery.	NCT02217956 [276]	[277]
**H + DT**	Non-small cell lung cancer	1 phase 2 phase	China	2015–2018	97	The median OS for the active arm was 9.4 months and for the control arm was 5.6 months. The median PFS for the active arm was 3.0 months and for the control arm was 1.85 months.	NCT02655913 [278]	[279,280]
**H + CH**	Breast cancer	1 phase 2 phase	United States	2006–2016	29 (Trial A = 18 Trial B = 11)	In Trial A: TTLP, PFS and OS was 4.9, 4.8, 9.0 months, respectively.In trial B: 6 of 11 patients had a clinically significant quality of life (QoL) improvement.	NCT00346229 (trial A) [281] NCT00826085 (trial B) [282]	[283]
**H + CH + S** **(HIPEC + CRS)**	Colorectal cancer	1 phase 2 phase	Italia	2006–2010	20	Median follow-up was 65.2 months in the HIPEC group and 34.5 months in the control group. 5-year overall survival (OS) was 81.3 % in the HIPEC group versus 70.0 % in the control group.	NCT02575859 [284]	[285]
**H + CH + S** **(HIPEC + CRS)**	Peritoneal carcinomatosis	1 phase 2 phase	France	2007–2011	18	—	NCT01226550 [286]	—
**H + R + S + CH**	Sarcoma	1 phase 2 phase	United States	1999–2007	15	—	NCT00093509 [287]	[288]
**H + CH + S + CH (HIPEC + S + CH)**	Ovarian carcinoma, fallopian tube carcinoma, primary peritoneal carcinoma	1 phase 2 phase	Belgium	2010–2015	19	The median follow-up was 30.9 and the median PFS was 33.2 months. The OS survival was not reached.	NCT01709487 [289]	[290]
**H + CH + S + CH (HIPEC + S + CH)**	Colorectal cancer	2 phase	China	2016–2020	100	—	NCT02830139 [291]	—
**H + CH + S + CH (HIPEC + S + CH)**	Stomach cancer	2 phase	China	2015–2020	100	—	NCT02528110 [292]	—
**H + CH + R**	Rectal cancer	2 phase	Germany	2012–2017	78	3-year evaluate for OS, DFS, LC and DC were 94%, 81%, 96%, and 87%, respectively. Higher cumulative temperatures associated with hyperthermia indicated stronger tumor regression in patients.	NCT02353858 [293]	[294].
**H + CH + S** **(HIPEC + LS)**	Gastrointestinal cancer	2 phase	United States	2014–2020	21	The median OS from the date of diagnosis of metastatic disease was 30.2 months. The median OS from the first laparoscopic HIPEC was 20.3 months.	NCT02092298 [295]	[296,297]
**H + R**	Prostate cancer	2 phase	United States	1997–2003	37	With a median follow-up of 70 months (18–110 months) 7-year OS was 94% with 61% of patient failure free.	NCT00003045 [298]	[299]
**H + CH +S** **(HIPEC + CRS)**	Desmoplastic small round cell tumor (DSRCT) sarcoma	2 phase	United States	2011–2018	22	The estimated median OS from the time of diagnosis was 58.44 months (for 20 patience).	NCT01277744 [300]	[301]
**CRS + HIPEC + EPIC**	Peritoneal carcinomatosis gastric cancer	2 phase	Sweden	2005–2009	18	The OS was 14.3 months for 8 patients who received entire treatment. The median OS for the CRS + HIPEC + EPIC group of patience was 10.2 months. 6 patients had macroscopically radical surgery (CC0) and for this subgroup OS was 19.1 months.	NCT01379482 [302]	[303]
**H + CH + S** **(HIPEC + CRS)**	Peritoneal carcinomatosis, colorectal cancer, appendiceal cancer peritoneal mesothelioma, pseudomyxoma peritonei, gastric cancer	2 phase	United States	2011–2020	51	—	NCT02040142 [304]	—
**H + CH + S** **(HIPEC + CRS)**	Adrenocortical carcinoma, peritoneal carcinomatosis	2 phase	United States	2013–2018	11	The median follow-up was 23 months. The median IP-PFS was 19 months. The median OS had not yet been reached.	NCT01833832 [305]	[306]
**H + CH + S + CH** **(IPHC + CRS + CH)**	Colorectal cancer	2 phase	United States	2002–2012	27	The median follow-up was 40.4 months. The median OS and PFS were 43.0 and 9.3 months, respectively.	NCT00310076 [307]	[308]
**H + CH**	Sarcoma	2 phase	United States	1996–2003	34	—	NCT00002974 [309]	—
**H + CH**	Melanoma	2 phase	United States	1995–2000	34	—	NCT00002973 [310]	—
**H + CH + S** **(HIPEC + CRS)**	Ovarian cancer	2 phase 3 phase	Republic of Korea	2010–2020	184 (HIPEC, 92; control, 92)	Two-year PFS was 43.2% and 43.5% and 5-year PFS was 20.9% and 16.0% in HIPEC and control group, respectively. Five-year OS was 51.0% and 49.4% in HIPEC and control group, respectively. In women who received NAC, the median PFS for HIPEC and control group were 20 and 19 months and the median OS for HIPEC and control group were 54 and 51 months, respectively. In the subgroup with NAC, 2-year PFS was 37.2% in HIPEC group and 29.5% in control group and 5-year OS was 47.9% in HIPEC group and 27.7% in control group. After 20 months in PFS and 30 months in OS.	NCT01091636 [311]	[312]
**H + B**	Cervical cancer	3 phase	Poland	2006–2009	224	Statistical differences were not observed for the distribution of early and late complications between the HT and non HT groups.	NCT01474356 [313]	[314]
**H + CH + S + CH** **(HIPEC + CRS + CH)**	Colorectal cancer primary peritoneal cavity cancer	3 phase	France	2008–2015	265	The median follow-up of was 63.8 months, the median OS was 41.7 months in the cytoreductive surgery plus HIPEC group and 41.2 months in the cytoreductive surgery group.	NCT00769405 [315]	[316]
**CRS + HIPEC**	Ovarian cancer	3 phase	Netherlands	2007–2017	242	The median OS was 45.7 months in the surgery-plus-HIPEC group and for surgery group of patience the median OS was 33.9 months.	NCT00426257 [317]	[318,319,320]
**H + CH**	Sarcoma	3 phase	Germany	1997–2012	340	Median follow-up was 11.3 years. Patients randomized to chemotherapy plus hyperthermia had prolonged survival rates compared with those randomized to neoadjuvant chemotherapy alone with 5-year survival of 62.7% vs 51.3%, respectively, and 10-year survival of 52.6% vs 42.7%.	NCT00003052 [321]	[322,323]
**H + B**	Cervical cancer, prostate cancer	N/A	United States	2009–2020	13	—	NCT00911079 [324]	[325]
**H + CH**	Bladder cancer	N/A	Turkey	2012–2017	44	In the intermediate- and high-risk groups, the recurrence free survival rates at the 24th month were 78.6% and 80% and the progression free survival rates were 92.6% and 76.7%, respectively.	NCT03694535 [326]	[327]
**PDT + TT**	Basal cell carcinoma	1 phase	United States	2015–2017	4	ORR showed 90% CR and 10% PR for the study.	NCT02639117 [328]	[329]
**PDT + CH**	Pancreatic cancer	1 phase	United States	2013–2018	12	The median follow-up of 10.5 months, PFS and OS were 2.6 months and 11.5 months, respectively.	NCT01770132 [330]	[331]
**PDT + S**	Non-small cell lung cancer	1 phase	United States	2014–2018	8	—	NCT01854684 [332]	—
**PDT + S**	Head and neck cancer	1 phase	United States	2006–2018	15	The clinical follow-up visits at 48 months showed OS of 10 patients and PFS of 7 patients. The primary objective was to determine the safety of HPPH-mediated intraoperative adjuvant PDT immediately after tumor resection and to determine the highest dose of laser light that can be safely used in treatment.	NCT00470496 [333]	[334]
**PDT + B**	Lung cancer	1 phase	United States	1993–2004	—	—	NCT00014066 [335]	—
**PDT + ER**	Early cancer in Barrett’s esophagus	2 phase	United States	2005–2012	73	—	NCT00217087 [336]	—
**PDT + S**	Malignant mesothelioma	2 phase	United States	1999–2010	12	—	NCT00054002 [337]	—
**PDT + S**	Non-melanoma Skin cancer	2 phase	United States	1993–2007	—	—	NCT00002963 [338]	—
**PDT + CH**	Perihilar cholangiocarcinoma	3 phase	Republic of Korea	2009–2013	43	Patients treated with combinatory therapy showed higher 1-year SR compared with the patients treated with PDT alone: 76.2% vs. 32% and median prolonged OS was 17 months vs. 8 months. Median PFS for combinatory treatment was 10 months and for patients with PDT alone was 2 months.	NCT00869635 [339]	[340]

Table 3 completed, interventional clinical trials involving hyperthermia and PDT in combination therapy in cancer patients. Information was retrieved from ClinicalTrials.gov (last assessed on 26 April 2021), by searching for ‘condition or disease: ‘cancer’, other terms: ‘hyperthermia’ or ‘PDT’, status recruitment: ‘completed’, study type: ‘interventional studies (Clinical Trials)’. Schema of combinatory treatments: H—hyperthermia, R—radiotherapy, CH—chemotherapy. PDT—photodynamic therapy, TT—target therapy, S—surgery, DT—drug therapy, B—brachytherapy, ER—endoscopic resection, LS—laparoscopic surgery, FRWBH—fever-range whole body hyperthermia, CRS—cytoreductive surgery, HIPEC—hyperthermic intraperitoneal chemotherapy, EPIC—early postoperative intraperitoneal chemotherapy, IT—immunotherapy. * (Local regional control (LRC), progression-free Survival (PFS), overall Survival (OS), time to local progression (TTLP), local control (LC), distant control (DC), disease-free survival (DFS), overall response rate (ORR), complete response (CR), partial response (PR), survival rate (SR), not applicable (N/A)).

### 4.1. The Synergistic Effect

The general idea of synergy is the interaction of various factors. Furthermore, in the context of cancer treatment, it involves cooperation of therapeutic methods. The difference between a synergistic and an additive action is that in synergy, the mechanism is based on mutual influence, while in the case of additivity, it involves adding individual methods and their effect. Synergistic effects should have a greater impact and should give better results than the sum of individual methods used in treatment—this is where the new treatment quality emerges. Considering the phenomenon of the synergistic effect, in the context of treatment it can be defined as strengthening the mutual effects of methods in cases where various treatment techniques are applied in a multi-module or combined approach. This approach is designed to strengthen the desired therapeutic results. These results arise only when we are dealing with a minimum of two factors, two substances, or—in the context of therapeutic treatment—two methods. Undoubtedly, the synergistic effect increases the effectiveness of the therapy. It is an important approach that stands behind combination therapy, since modular operation allows the initiation of a cascade, which has a positive effect on the effectiveness of treatment. Of course, the synergistic effect does not always have to be positive. Unfortunately site effects can get enhanced as well. Nevertheless, the aim of designing combination treatment is to enhance anti-cancer modalities and, at the same time, reduce harmful effects of the treatment.

When achieving a synergistic effect by combining different therapies, we can attain inhibition of cancer cell proliferation, destruction of the cytoskeleton and, thus, increase the chance of apoptosis. It also results in damaging blood vessels, reducing the resistance of cancer cells to chemotherapy or radiation. In addition, it strengthens the mechanisms of action of the drugs used. The concept of synergy is quite often an issue when combining chemotherapeutics with different mechanisms of action. Cisplatin, gemcitabine, or vinorelbine can also be combined with proteasome inhibitors in the treatment of lung cancer. According to scientific research, we can achieve quite good results using the above method. Additionally, interesting effects can be obtained by using abraxane and gemcitabine. The scheme of action using these two drugs was as follows: first nab-paclitaxel was injected into the mice, and then gemcitabine. The time interval between the injections of these drugs was 2 h. This treatment procedure was performed in six cycles. In this experiment, radio frequency electromagnetic radiation (called radio frequency—RF) was used. The use of RF allowed inducing hyperthermia in the tumor area. The experiment showed the effectiveness of such a treatment. A reduction in the size of the PANC-1 orthotopic tumors was observed [341]. On the other hand, murine 4T1 tumors with a low degree of vascularization were characterized by the greatest functional increase in the vascular system after the use of hyperthermia, which, according to the authors, may be beneficial in a treatment involving the use of chemotherapy [342]. On the other hand, with regard to PDT, hyperthermia can cause tumor blood vessel damage that would affect angiogenesis and, like PDT, damage cell membranes, which could allow the occurrence of a synergistic effect between these two approaches [343]. 

When discussing the results of synergistic effects, it is usually worth emphasizing that, when treating cancer with the use of different methods, the overall idea is to act sequentially. One method should complement and strengthen the previously used one. What is important is that this approach allows using a smaller dose of a drug, a chemotherapy agent, and a photosensitizer or radiation, and it can reduce the side effects of the therapy. It is worth emphasizing that, when synergistic effects occur, they increase irreversible damage (e.g., by affecting cell repair abilities) and radiosensitivity, there is also an increase in regard to the sub-lethal damage. Different mechanisms can be alerted too, e.g., one therapy component can first induce immune response, then it can act as an anti-angiogenic agent, and then–target cancer cells. In addition, inhibition of repair of any damage may occur, as well as an increase in the sensitivity of hypoxic cancer cells. The increase in the concentration of chemotherapeutics inside the tumor is also significant.

### 4.2. Attempts to Treat Tumors with PDT and Hyperthermia 

Scientific research, publications, and scientific articles describe treatment by photodynamic therapy and hyperthermia on cancers, such as skin cancers (e.g., melanomas, ocular melanoma, colon, bladder, esophageal, breast, head and neck, lung cancers) [344,345]. It is difficult to find information about research connected with treatment with the use of PDT and hyperthermia in the context of treating cancers associated with the cardiovascular system and the circulatory system, such as leukemia. Therefore, knowing the specifics and general principle of these two methods, the cancers of choice are either on the surface of the skin, i.e., in this case any cancerous lesions of a melanoma nature, or inside the body. PDT, as well as hyperthermia, thanks to the use of probes, allow for quite precise access to a given area that is to be treated [227,346,347] In addition, these methods are also used to treat organs, such as liver, lungs, or pancreas that are relevant to our bodies, and within which, tumorous lesions were detected in locations where surgery is not an option due to the fact that it is quite easy to damage such extremely important structures. Due to the fact that PDT and hyperthermia are treated as complementary therapies, these methods are also used in the context of solid tumors and primary tumors, but also in the context of treating malignant tumors that have a tendency to metastasis. Therefore, it seems that attempts to treat cancers with photodynamic therapy or hyperthermia are practiced on most common cancers [347,348].

## 5. Combination Therapy’s Effect on Drug Uptake and Delivery

Hyperthermia and photodynamic therapy play a significant role in drug delivery and absorption during cancer treatment. Hyperthermia can possibly change the level of drug delivery. It can break and unblock certain restrictions related to the tumor location or structure. Moreover, it can alter the cytoskeleton itself. Temporary damage to the cytoskeleton causes deformation of cells and vessels surrounding the tumor, which in turn facilitates the drug transport [349]. It is also worth emphasizing that hyperthermia increases the permeability of blood vessels [350], which significantly affects the delivery of drugs to the area of the neoplastic lesion and improves the effectiveness of the drugs used. The efficacy of drug delivery in cancer treatment therapies is deliberately improving. Targeted treatment [351], where innovative and modern carriers are used to make the uptake, absorption, and transport of drugs within the neoplastic lesion to be much more effective, seems to be a good approach. This approach allows getting rid of many negative and undesirable side effects associated with drug distribution and toxicity. PDT combined with other therapies, for example with chemotherapy, allows the use of a much lower dose of the chemotherapeutic in most of the cases [352]. The problem with drug resistance is significantly reduced, and once again we are dealing with a reduced toxicity. Due to the synergistic effects caused by the use of these therapeutic methods, the dose required to obtain the desired therapeutic effects can be lowered. Any modification of drugs and their delivery, for example by liposomes during hyperthermia or PDT, significantly influences drug absorption as well. This approach ensures efficient delivery and bounding the drug in the tumor region, which significantly affects cancer cells and causes cell damage [353]. A therapy involving the use of the thermal aspect may also increase the rate of drug release in a given area where the temperature will rise significantly [354]. There is an important issue in the context of drug transport related to thermoablation. It is interesting that under the influence of high temperature, which can be induced by hyperthermia, it is possible to increase the proliferation of cancer cells by activating various pro-tumorigenic factors [355]. Normalization as part of PDT is also worth mentioning. The normalization effect is based on the fact that there is a temporary normalization of blood vessels, both in terms of their structure and function [356]. This creates an individual therapeutic window through which a better therapeutic result can be obtained. In the context of normalization of vessels with PDT, there is also an issue related to the delivery of certain substances to the vicinity of neoplastic tissue. These substances are supposed to work against excessive angiogenesis. Anti-VEGF angiogenesis inhibitors are designed to restore some degree of regularity in damaged, deformed, and leaky blood vessels, and to increase the oxygenation of the tissue surrounding the tumor [357]. The occlusion of blood vessels that occurs with the use of these therapies is extremely important [358]. Under the influence of the increased temperature, the neoplastic tissue increases in volume due to hyperthermia. As a result, the pressure inside the tumor changes. When pressure in the tumor area is lowered, a leakage occurs. As a result, the pressure around the tissue rises. Moreover, a properly selected combination therapy should ideally ensure that the drug is first delivered under the influence of the generated heat, and after the occlusion of the blood vessels, the therapeutic substance is closed inside the tumor due to the vascular pruning mechanism. It is worth emphasizing that attempts are being made to include nanoparticles in the diagnosis, for example quantum dots, and in the treatment of cancer. In particular, they enable more effective detection of early lesions and early stages of cancer. In addition, the use of nanogold contributes to the drug distribution if a construct with nanoparticles is used in the tumor area. Besides, the treatment effectiveness can be increased by combining nanoparticles with antibodies. This is a fairly promising approach. Nanoparticles can be used in many therapies. When it comes to combining nanoparticles with chemotherapy, this approach is primarily intended to increase the efficiency of the delivery of a chemotherapeutic drug specifically to the tumor area [359]. As a result, the effectiveness of treatment may increase, and due to the fact that we act specifically, it is possible to reduce the side effects associated with the toxicity of chemotherapeutics. When it comes to radiation therapy and combining this method of cancer treatment with nanoparticles, then the task of nanoparticles is to sensitize cancer cells. For example, gold nanoparticles can increase the effectiveness of radiation therapy due to the fact that the production of free radicals increases in a specific place, therefore the area becomes more oxidized, the level of hypoxia decreases, and cells are more sensitive to a given dose of ionizing radiation. Nanoparticles have their place in PDT as well [360]. Photosensitizer carriers have been created, which are able to reach the tumor area much more effectively than regular carriers. In the context of hyperthermia, it is quite common to use nanoparticles that exhibit magnetic properties. As a result, they are used as an alternative source of energy in this therapy [361]. Gold nanoparticles in the context of hyperthermia are also used in thermoablation. This phenomenon allows the destruction of tumors. In general, nanotechnology is beginning to play a significant and important role in the context of cancer treatment, and combining nanoparticles with methods that are currently used in treatment is now much more common [362]. This gives promising results for the future and thanks to the use of nanoparticles, it is possible to treat patients more effectively and minimize the side effects of therapy. Considering all of these aspects, it can be clearly stated that both hyperthermia and photodynamic therapy are beneficial in terms of drug delivery. The use of the above-mentioned therapeutic methods makes the therapeutic effect more beneficial, and the issue of drug delivery and better absorption can be improved.

## 6. Photodynamic Therapy and Hyperthermia in Combination Treatment

PDT and hyperthermia can also be combined. Particularly, hyperthermia could solve the issue of hypoxia for PDT. In studies on rat tumors, Kelleher et al. demonstrated that combined treatment, which consists of conducting photodynamic therapy based on aminolevulinic acid (ALA-PDT) simultaneously with local hyperthermia at 43 °C, is more effective than the sum of its components [363]. In addition, vascular collapse and flow stasis were shown to be a key element in tumor elimination in combined hyperthermia and chlorophyll photodynamic therapy (Bchl-ser-PDT), resulting in lower tumor oxygenation and a switch from oxidative to glycolytic glucose turnover [364].

However, in PDT, widely recognized as a highly effective precision therapy, the key challenge remains to refine it for use in hypoxic tumors. One approach to solving the problem of hypoxia may be the combination of PDT with hyperthermia. Li et al. demonstrated against hypoxic tumors the use of polymer vesicles that are capable of deeply penetrating the tumor and at the same time providing oxygen delivery after irradiation with light. Namely, after exposure to light, a thermal effect is induced, which can decompose hydrogen peroxide into oxygen, and then after irradiation at 660 nm, the vesicles are quickly destabilized by splitting the copolymer with singlet oxygen under the influence of light irradiation, which allows the release of photoactive poly(amidoamine) dendrimer conjugating chlorin e6/cypate (CC-PAMAM) [365].

Moreover, Kurokawa et al. also showed that treatment with hyperthermia (42 °C) can increase the production of mitochondrial reactive oxygen species (mitROS), thereby enhancing the effects of PDT in cancer cells. The mechanism of this phenomenon was most likely due to both the increase in the expression level of the heme carrier protein-1 (HCP-1) and the decrease in the level of the ABCG2 transporter by mitROS [206].

## 7. Proposed Combinations That Are Currently Used in Multimodal Cancer Treatment

Radiotherapy, chemotherapy, and surgical intervention are reliable options for treating cancer [366]. Currently, multi-module treatment is used quite commonly to treat cancer: chemotherapy and radiotherapy are used together very often. In addition, attempts are being made to enhance therapeutic effects by improving the mechanisms on which standard treatments are based. These attempts focus on aspects related to delivery of drugs, photosensitizers, or chemotherapeutic agents in order to minimize side effects and to act selectively. Therefore, drugs can be delivered using liposomes, and chemotherapeutics with a different spectrum of activity can be used to enhance therapeutic effects. In addition, research using gene therapy is being carried out, and new drug are being designed. New mechanisms have been developed to achieve better results without damaging healthy structures, which is why, for example, a radiation-based gamma knife or cyber knife are used.

### 7.1. Radiotherapy 

Radiotherapy is an important part of cancer treatment, and its main goal is to deprive cancer cells of their proliferation potential. Radiation is a physical factor that stores energy in the cells of the tissues it passes through, and then this deposited energy can kill cancer cells or cause genetic changes that lead to cancer cell death [367]. The main mechanism of killing cells by high-energy radiation involves damaging their genetic material of DNA, thereby blocking their ability to further divide and proliferate [367,368]. The goal of improving radiation therapy is to maximize the radiation dose to cancer cells while, at the same time, minimizing the exposure of healthy cells that are adjacent to cancer cells or are exposed to radiation [367]. Radiotherapy is also used in conjunction with other treatments, such as surgery, chemotherapy, or immunotherapy. 

Currently, brachytherapy and teleradiotherapy are commonly used. Brachytherapy involves treatment with the use of a radiation source that must be as close to the tumor as possible. The mechanism of brachytherapy is based on the use of radiation in direct contact with the tumor. However, when it comes to teleradiotherapy, the source of radiation is located at a certain distance from the tissue within the area of the cancer. In the context of radiotherapy, the so-called 5R principle that occurs during radiotherapy: redistribution, repopulation, reoxygenation, repair, and radiosensitivity is worth highlighting [369]. Radiation can be classified as ionizing radiation (e.g., X and gamma radiation) or particle radiation with electrons, protons, alpha particles, and neutrons. The biological mechanism of action depends on the type of radiation (e.g., relevance of linear transfer energy and cell damage). On the other hand, cancerous tissue parameters can provoke effects, such as the increase in sensitivity of cancer cells to ionizing radiation if the activity during the division of cancer cells is also greater. However, a higher level of differentiation makes cells less sensitive to radiation. The level of tumor oxygenation is also an important factor. When tumor is more hypoxic, the cancer cells are less sensitive to radiation (oxygen enhancement ratio is 2.5–3 times bigger for a well oxygenized tissue). In context of radiotherapy, there is a very interesting strategy to treat malignant eye tumors—protonotherapy. This technique uses Bragg’s pick to deposit energy into the tumor, and the healthy tissue around the tumor is kept safe at the same time [370]. It was shown that proton radiation can inhibit the metastatic potential of primary cancer cells. 

In the outcome of radiotherapy methods and mechanisms, temporary inhibition of tumor growth or growth retardation and tumor regression may take place. There are many factors that have to be considered, although these mechanisms can be influenced by the duration of the cell cycle, the size of the cell growth fraction or the rate of cell loss. 

From the perspective of potential needs for combination treatment involving the use of radiotherapy, a few factors should be taken into consideration. Regarding the above-mentioned cell sensitivity to radiation (e.g., cell cycle, oxygen enhancement), the main need is to overcome radioresistance. For example, melanomas are quite resistant to radiation therapy due to their melanin accumulation (accumulation of pigment that acts as a radioprotector). Additionally, cancer cells that contain melanin are hypoxic and, therefore, more resistant to low LET radiation. A high level of cell differentiation also contributes to this resistance [371]. In the treatment of melanoma where melanin is present, its role is to scavenge free radicals. This mechanism allows putting this pigment in the radioprotector category. It protects melanin against the impact of ionizing radiation, for example X-radiation. What is worth emphasizing is that this pigment increases radiation resistance by inactivating free radicals that are formed during the course of radiation. A well-applied complementary treatment can increase melanoma sensitiveness to ROS. 

The limitations of radiotherapy, such as insufficient tissue oxygenation can be overcome by using complementary therapies. One such technique is hyperthermia, which is usually a complementary method that involves heating the tumor in order to inhibit the proliferation of cancer cells, destroy them, or make them sensitive to various treatments, including radiation therapy. The combination of hyperthermia and radiotherapy shows a synergistic effect and enhances the killing effect on cancer cells, especially those in the S phase of the cell cycle, which are usually resistant to radiation when applied alone [19]. The synergistic effect of heat and radiation is defined as the thermal enhancement ratio (TER), which defines the magnitude of thermal hypersensitivity to radiation as the survival fraction quotient after irradiation alone and in combination with hyperthermia [372]. The effects of hyperthermia include, but are not limited to, inhibition of the repair of radiation-induced DNA damage, thereby increasing the cytotoxic effect of radiotherapy [372]. Moreover, by reducing the metabolic activity of target cells, heat reduces the oxygen demand of the tumor and increases the oxygenation of the tumor tissue as well, making hyperthermia one of the most powerful radiosensitizers available [373].

Another approach to overcome the limitations of radiation therapy is to combine it with photodynamic therapy. Photodynamic therapy uses photosensitizers that are activated by visible or near-infrared light and transfer energy to molecular oxygen, thus generating reactive oxygen species [373]. Under certain conditions, some photosensitizers can act as radiosensitizers. Combining radiation therapy with photodynamic therapy, i.e., using ionizing radiation tissue penetration and photodynamic therapy can reduce penetration depth problems and can allow radiation dose reduction without decreased clinical efficacy [374], while minimizing damage to healthy tissues. Moreover, the combination of an appropriate photosensitizer with radiation therapy can lead to a significant increase in the cytotoxic and apoptotic death of cancer cells [375]. The combination of photodynamic therapy and radiation therapy, the primary goal of which is to kill cells through nuclear DNA damage, offers the possibility of synergism in killing cells, as photodynamic therapy does also induce several types of DNA damage [376]. In addition, photodynamic therapy does also improve the immune system’s response by inducing inflammation and an immune response against cancer cells. Potential mechanisms of immune stimulation by photodynamic therapy include an acute inflammatory response that can enhance tumor antigen presentation to activate dendritic cells and to guide them to regional and peripheral lymph nodes, ultimately stimulating cytotoxic T lymphocytes and NK cells, accompanied by the formation of immune memory and growth suppression tumor in the future [255]. 

### 7.2. Chemotherapy 

Chemotherapy is a very broad category of anti-cancer treatment. The induction of apoptosis and the inhibition of mitosis as well as the cell cycle disorder are caused by the use of chemotherapeutic agents. Cytostatic drugs can be grouped into alkylating agents, alkaloids, antibiotics, and antimetabolites. Inhibitor of tyrosine kinases are also very important novel drugs. They affect cell proliferation by targeting cellular DNA or RNA and metabolism with antimetabolites acting on purine or pyrimidine metabolic enzymes, while alkaloids act on the cytoskeleton and mitosis [377]. One of the main problem with chemotherapy is the effective, safe, and selective drug delivery. Chemotherapy is associated with the presence of side effects that include immediate signs of toxicity (effects can be seen on skin and hair, bone marrow and blood, gastrointestinal tract and kidneys, etc.) as well as late signs of chronic toxicity (drug resistance, carcinogenicity) [377]. In order to increase the effectiveness of chemotherapy and reduce side effects, a combination of various drugs with different mechanisms of action is used. Moreover, when several chemotherapeutic agents are used, drug resistance could be counteracted. Overall results of chemotherapy are also improved due to the use of combinations of drugs that do not have overlapping toxicities. Then, we can increase the dose of drugs in the tumor without fearing undesirable negative side effects of the therapy. This approach is used very often to reduce side effects since increasing the dose of only one drug can cause toxicity; therefore, it seems that combining drugs from different groups is less toxic and more effective due to the higher therapeutic dose applied. The same happens when we use drugs that have different mechanisms of action. In this case, cells that are insensitive to one drug are already sensitive to the other drug from this combination. Treating drug-resistant cancers is a significant problem not only in chemotherapy. This mechanism of drug resistance may be influenced by the fact that the chemotherapeutic agent alone is not able to reach the inside of the tumor directly. Still, other challenges face this approach as well, such as improving drug perfusion and increasing the accumulation of the therapeutic compound in the tumor. Hyperthermia, which causes changes in cells and their surroundings due to high temperature, seems to be a good complementary technique. In addition to direct ablation of cancer cells, elevated temperatures can also trigger drug release, especially for heat-sensitive carriers [378]. It is also known that raising the temperature from 37 °C to 43 °C such increases the permeability of the cell membrane, accelerating the absorption of nanoparticles, and it may increase the interaction of carriers with the cell membrane as well [379]. In particular, energy-dependent pathways, such as clathrin- and caveolae-mediated endocytosis are involved in increasing the permeability of the cell membrane at high temperature, thus increasing the internalization of drugs and carriers [378,379]. Hyperthermia also affects blood flow and, hence, changes the drug distribution. Certainly, the heat-induced change in blood flow in tumors differs from the one that occurs in regular tissues because the tumor vasculature is less able to dissipate heat and is more susceptible to damage during hyperthermia treatment. However, it is worth noting that mild heat increases blood flow in the tumor, allowing chemotherapy to have a greater effect on cancer cells [373]; thus, in some types of tumors, blood flow increases when heated to relatively low temperatures. On the other hand, higher temperatures (43 °C or 44 °C) result in stronger and longer-lasting vascular closure [380]. 

In addition to allowing more efficient drug delivery, heat can also modify the cytotoxicity of many chemotherapeutic agents. In many cases, synergism can be seen as a continual change in the rate at which the drug kills cells as temperature increases. The cytotoxicity of most alkylating agents, platinum compounds, and also nitrosoureas increases linearly with the temperature increase, typically by thermal enhancement, including increased alkylation rate constants, increased drug absorption, and inhibition of repair of lethal or sublethal drug-induced damage [381]. 

Additionally, hyperthermia may increase the killing effect on tumor cells located at the hypoxic centers of tumors that are relatively resistant to chemotherapy due to poor drug delivery. In addition, some chemotherapy drugs also require oxygen to generate free radicals in order to induce tumor cytotoxicity. It is known that elevated temperatures can increase the rate of biochemical reactions, increasing cellular metabolism, which can result in an increased oxidative stress. The level of reactive oxygen species may increase after exposure to elevated temperatures, possibly due to dysfunction of the mitochondrial respiratory chain or by increased activity of the enzymes NADPH oxidase and xanthine oxidase [382]. 

### 7.3. Surgical Intervention

Surgery is one of the three most popular treatments for cancer today, and also the oldest one. It plays the most important role in the treatment of cancer. Its application ranges from the diagnosis of lesions, i.e., taking biopsies for diagnosis, through reducing the tumor mass, to radical treatment in order to completely excise the lesion together with a margin of healthy tissues. Complete tumor excision with cutting out a healthy tissue margin is the most effective form of treatment in the case of early stage neoplasms, especially when there are no metastases yet. Such achievement of a sufficiently negative margin during oncological surgery minimizes the risk of adverse treatment outcomes and recurrence of the disease [383]. In some cases, the unfavorable location of the tumor or the presence of disseminated metastases can make it impossible to remove the tumor. Hence, it is not always possible to obtain the necessary margin, in particular in case of surgery in the vascular system area, in case of other critical areas, and in case of tumor involvement in adjacent tissues. This problem applies to such neoplasms as hepatocellular carcinoma, pancreatic ductal adenocarcinoma (PDAC), neuroblastoma, or neuroendocrine tumors of the gastrointestinal tract. Involvement of a blood vessel can sometimes be resolved by surgical resection and reconstruction of the affected vessel, such as a vein or artery, but these procedures are associated with an increased risk for the patient, especially in the case of arteries [384,385].

In conclusion, failure to obtain adequate surgical margins increases both the surgical and oncological risk of poor prognosis, which is usually the case when tumors have invaded large blood vessels [385]. If it is impossible to remove the entire neoplastic lesion, the combination of surgery and radiation therapy is often used. Postoperative radiotherapy reduces the risk of cancer recurrence and also helps destroy any remaining cancer cells, especially when only the tumor was removed and only a small amount of regular tissue around it was removed, or a margin is left that is positive for cancer cells. Indications for postoperative radiotherapy include not only insufficient resection margin, but also uncertain radicality of resection, infiltration of tissue with diffuse cancer foci, and low tumor differentiation. Therefore, in most cases, only combinations of treatments such as surgery combined with radiation therapy are usually the only way to destroy cancer cells [373]. 

Returning to the issue of tumor areas that cannot be surgically excised due to the fact that their location is too close to inoperable vessel, there is one interesting complementary method: the use of local, mild hyperthermia. By applying uniform and gentle heating, we can destroy tumors that surround the vessels and, at the same time, protect these sensitive structures from damage [385]. It has been shown (on the example of pancreatic ductal adenocarcinoma) that an increase in temperature in the range of 41–46 °C leads to killing cancer cells, including the elimination of cancer stem cells, as well as changes in the proteomic profiles of cancer cells, with simultaneous protection of regular cells [385].

Another method that is complementary to surgery is photodynamic therapy, which is based on the complex cell killing phenomenon resulting from the interaction of a chemical compound (photosensitizer), light, and oxygen. Photodynamic therapy is recognized as a safe and effective method, and this is why it plays a unique role in the treatment of cancer with its targeting precision. It does not damage healthy structures surrounding the treated lesions, and it is applied in the treatment of cancerous tumors with limited access [386]. In systemic photodynamic therapy there is a wide distribution of photosensitizer, but there is also a higher potential for accumulation in cancerous tissues than in healthy tissues, and multi-site deep light, which can be performed intraoperatively in combination with standard or minimal surgical access [386]. Consequently, by using the preferential accumulation of photosensitizers in cancer cells with the appropriate selection of irradiation, it is possible to eliminate the remaining tumor fragments that cannot be surgically removed. Photodynamic therapy may be used in some cases before surgery as a neoadjuvant therapy to alleviate cancer. 

In addition to destroying cancer cells, photodynamic therapy is closely related to the fluorescence phenomenon used in photodiagnostics to detect lesions. As noted, the optical properties of healthy and abnormal tissues in the ultraviolet and visible spectrum differ from each other. Due to endogenous chromophores, some tissues exhibit a characteristic fluorescence emission band that changes when the disease process occurs, changing these chromophore components [386]. Thus, using the appropriate wavelength, an image can be obtained that is used as the basis of autofluorescence photodiagnosis [386,387]. This method may be helpful in determining the optimal biopsy site for histological diagnosis. In addition to using endogenous chromophores, photosensitizers can be used as exogenous fluorophores to enhance fluorescence [386]. The use of such enhanced photodiagnostics is especially valuable in surgical operations where it can indicate residual neoplastic infiltration, which may be present at the margin of resection. In addition, it can be used in brain surgeries, in the removal of tumor remains that are invisible to the naked eye or with the use of microscopic surgical instruments [386,388].

## 8. Hypoxia as a Treatment Imitation Factor

The phenomenon of hypoxia is defined as the state of O_2_ partial pressure reduced below critical thresholds specific for particular tissue, or from a biochemical point of view as O_2_-restricted electron transport [389]. Low oxygen tension often occurs in cancer cells due to several mechanisms, e.g., poor angiogenesis and/or increased oxygen consumption, pathological vasculature, and anemia. Hypoxia is an oxygen tension in the range from <0.01% (anoxia) to 5%, and it can be chronic, acute, or cyclic, with different effects on cancer cells [390]. From a radiobiology perspective, hypoxic fraction (percentage of hypoxic volume inside the tumor) defined by pO_2_ < 10 mm Hg is crucial for the therapy response [391]. 

Hypoxia makes cancer cells more aggressive, it contributes to their resistance to all kinds of treatment. The amount of oxygen in a tumor depends on the tumor stage and size, and on the cell metabolism as well. Signaling pathways associated with tumor hypoxia is usually induced by HIF induced hypoxia factors. Overexpression of HIF-1α and HIF-2α causes an increase in angiogenesis, aggressiveness, and resistance to treatment [392]. Low levels of oxygen are usually found in tumors. It may be caused by impairment, improper structure of blood vessels. Hypoxia of tumors is a serious challenge in cancer therapy, and hypoxic tumors are often much more difficult to treat than the well-oxygenated tumors. 

It was shown that that the occurrence of hypoxia vary independently on tumor size and type. This is definitely a factor that contributes to increased resistance to treatment. Tumor-building cancer stem cells (CSCs) are recognized as potential initiators of induction, progression, and increased metastatic capacity. In addition, cancer stem cells make the tumor much more heterogeneous, which definitely affects its resistance to any therapies that are aimed directly at tumor destruction. At this point, hypoxic niche could determine the phenomenon of hypoxia. Hypoxia itself is believed to be a factor that actually contributes to the phenotype of stem cells. In hypoxic niche, stem cells become carcinogenic cells and cause further metastasis. Hypoxic cells are much less susceptible to treatment, and due to the presence of a chaotic network of blood vessels within the tumor, the ability to accumulate chemotherapeutic agents is significantly reduced [393].

Due to tumor hypoxia, cancer therapy usually becomes less effective. Unfortunately, this situation leads to an increase in cancer spreading. Hypoxic cells are definitely more aggressive as compared to normoxic ones. Such as hypoxic cell are undernourished, and are well adjusted to harsh microenvironment which created hypoxic niche. One of examples of how different hypoxic environment can be acidic pH correlated with low pO_2_ [394]. When the tumor area is heated, such cells should be more susceptible to damage and death than cells that proliferate properly and with regular metabolism. It is worth emphasizing that if there is a necrosis around the tumor, shrinking of cancer cells can be observed in most cases. On the other hand, cancer stem cells can survive in such an environment and can be responsible for future cancer recurrence. Hyperthermia and the associated increase in temperature in the tumor region affect cancer cells. The mechanism of thermoregulation is becoming handicapped and there is no efficient heat dissipation due to the damage to blood vessels [393]. 

As we know, there is an area of low oxygenation within a tumor. This condition lowers treatment options as the cells are highly resistant to radiation. In order to counteract this phenomenon, the availability of oxygen within the tumor should be increased. Hyperthermia can be used for this purpose. It will increase blood flow through the blood vessels that entwine the tumor and supply it with nutrients, drugs, and oxygen. Increasing blood flow through the vessels should definitely improve the degree of oxygenation of the tumor itself and the surrounding tissues. Thanks to this approach, the state of hypoxia can be reduced, at least for a while, which will allow sensitizing cells to radiation in a given time window, and in the case of DNA damage, repair mechanisms will be weakened [395].

In general, hypoxic regions are poorly perfused, which has an influence on the drug accumulation potential. It is postulated that one of the reasons of hypoxic cells resistance is related to sublethal chemotherapeutic drug concentration. As a result, cancer cells gain abilities to remove drugs due to the presence of ABC transporters, and become more resistant to treatment. When there is a hypoxia, regulation of ABCG2 expression increases, and this allows a survival of a larger fraction of cells since the photosensitizer accumulates less intensively in the tumor area. This makes photodynamic therapy less effective [364,396]. Additionally, cells in hypoxic fractions of tumors are very often in G0 phase in the cell cycle, which makes them more resistant to chemotherapeutic drugs designed e.g., to stop DNA synthesis. These issues can affect localization of a photosensitizer that can accumulate in a manner dependent on its distance from blood vessels. However, researchers are trying to target drugs to hypoxic regions with two mechanisms: (a) active targeting–when the drug has affinity for molecular targets, like GLUT-1, and (b) passive targeting–when the drug has a better potential to localize in hypoxic fraction due to e.g., pH, lipophilicity, or size. Passive targeting in the time scale is a very interesting option for a photosensitizer–it was shown that a longer time interval between PS injection and illumination allows improving PS localization in poor perfused areas [21]. 

Cells susceptible to chemotherapy are mainly the types of cells that are characterized by a relatively fast rate of proliferation, which in turn distinguishes hypoxic cells that usually have a much slower degree of proliferation, which makes chemotherapeutic agents unable to act on slowly dividing cells. Drug resistance can occur as a result of stopping the cell cycle. Changing the state and lowering the rate of cell metabolism as well as increasing the rate of DNA replication also play an important role. Additionally, hypoxia may contribute to autophagy, which can cause both pro-apoptosis and survival of cancer cells. It has also been reported that the autophagy resulting from hypoxia is associated with the occurrence of drug resistance [397]. The autophagy process allows degrading damaged cell components, it is a kind of response before apoptosis that would occur due to the use of chemotherapeutic agents [398]. The use of, e.g., cisplatin allows for the stimulation of a defense response, as a result of which the autophagy process is initiated through the induction and regulation of AMPK and mTOR [399]. Hypoxia can also be caused by surgery in the area where the tumor was resected. Oncological surgery increases the ability of neoplastic cells to migrate and metastasize, and to relapse the tumor to the same place [400].

Research is currently underway to improve PDT where high levels of cancer hypoxia are involved. In PDT, hypoxia is clearly a serious problem, because the lack of oxygen in cancer cells leads to much less ROS production. To achieve an effective tumor response under hypoxic conditions, combination therapy would need to be optimized to promote some form of tumor reoxygenation [343]. The role of Hypoxia Inducible Factor’s (HIF’s) effect on tumor resistance is also significant. Low oxygen partial pressure can induce pleiotropic HIFs-related signaling cascade and regulate angiogenesis, apoptosis, metastasis, or cellular metabolism. After PDT, the hypoxia is very often observed inside the tumor, for a shorter or a longer period of time [21]. It needs to be highlighted that even very short hypoxic periods can provoke HIFs and enhanced angiogenesis. One of combination treatment strategies is the inhibition of VEGF after PDT. To summarize, in the context of photodynamic therapy, there are some photosensitizers that affect hypoxic cells. They selectively increase the sensitivity to radiation only in poorly oxygenated cells. It is also worth noticing that the destruction of the tumor with X-rays is often hampered by the existence of hypoxic cells that are very resistant to radiation and drugs. What is worth emphasizing in this context is that hypoxia in a tumor is often a local phenomenon. It is suggested that determination of the hypoxia area location in the tumor may allow the exposure of this area to a higher dose of radiation, which is to increase the therapeutic index [391]. 

The hypoxia makes cancer cells resistant to radiation, allows cancer cells to survive in a tumor’s microenvironment with a very low degree of oxygenation. Thanks to hypoxia, cells have an increased potential to inhibit damage in genetic material. When cells are in a hypoxic state, post-translational modifications of histones and DNA methylation may occur [401]. As it is known, during radiotherapy there is direct and indirect damage to DNA, and if there is some compensation and no damage to the genetic material due to hypoxia, the radiotherapy is not effective. In addition, the resistance to oxidative stress increases in hypoxia, which makes radiotherapy ineffective. Reducing the degree of radiation sensitivity is associated mainly with HIF-1α, which seems to play a significant role in the degree of tumor sensitivity to radiotherapy [402].

In this case, the use of fractionated radiotherapy to effectively treat oxygenated external parts of the tumor is one of the available options. This approach leads to reoxygenation of previously hypoxic tumor cells, which are still viable, and at this point, radiation therapy can be combined with PDT in order to effectively treat previously hypoxic parts of the tumor [343]. As already mentioned, another option would be to combine PDT with hyperthermia, which has been shown to target the vasculature of the tumor, which can then initiate tumor angiogenesis, leading to reoxygenation of tumor tissue [403,404]. However, this solution should be studied carefully due to the fact that tumor angiogenesis can lead to tumor growth. Hypoxia definitely has a negative effect on cancer treatment and reduces the effectiveness of all methods, and it is associated with a faster development of the disease. Undoubtedly, the impact of the hypoxic tumor environment on the effectiveness of combination therapy including PDT is also significant, and this should be taken into consideration when assessing different treatment options.

## 9. Conclusions

Due to cancer heterogeneity and the complexity of the disease, the proposed treatment procedure should be individualized. The type of cancer, specific biomarkers, stage of disease, and coexisting conditions are just some elements of a long list of factors that should be taken into consideration when planning the treatment. Choosing the therapy always involves finding a balance between fighting with cancer cells and minimizing side effects. Combination treatment is one of the most promising ways to reduce side effects, to focus on destroying cancer cells, and to stimulate anti-cancer immunity (Figure 2). In this context, photodynamic therapy and hyperthermia treatments appear as perfect candidates for complementary therapies. First, both of them can be site-specific and directly kill cancer cells by inducing apoptosis, necrosis, and different types of cell death. Second, both methods can have a significant influence on tumor perfusion and vessel structure in a dose-dependent manner in order to achieve: (a) vessels pruning, flow shut down, and severe hypoxia (for heat ablation and vascular targeted PDT); (b) stimulation of blood perfusion that can lead to good tissue perfusion and more effective drug accumulation inside the tumor (mild hyperthermia, cellular targeted PDT). Finally, hyperthermia and PDT can not only induce a strong immune response locally inside the tumor (inflammation), but can also trigger a systemic response against cancer. It was shown that PDT and hyperthermia alone can be effective against particular types of cancer, just like chemotherapy, immunotherapy, radiotherapy, and surgery. However, based on current statistics, 1 of 3 cancer patients is uncurable. Scientists are constantly developing new treatment options, but maybe more attention should be paid to bringing out a new quality from what we already know? Clinical experiences indicate that combination treatment is more than just a sum of basic therapies; the optimized multi-module treatment creates a new quality to help find a cure. The implementation of this type of treatment requires, above all, a qualified group of people and specialized equipment. For these therapies to be effective, it is good to target the therapeutic window, thanks to which the results will be more effective. Most importantly, to be able to start such treatment at all, it is necessary to thoroughly understand not only the mechanism of action of the therapy, but also the kinetics of the action of nanodrugs. All of this will translate into higher treatment success.

## Figures and Tables

**Figure 1 pharmaceutics-13-01147-f001:**
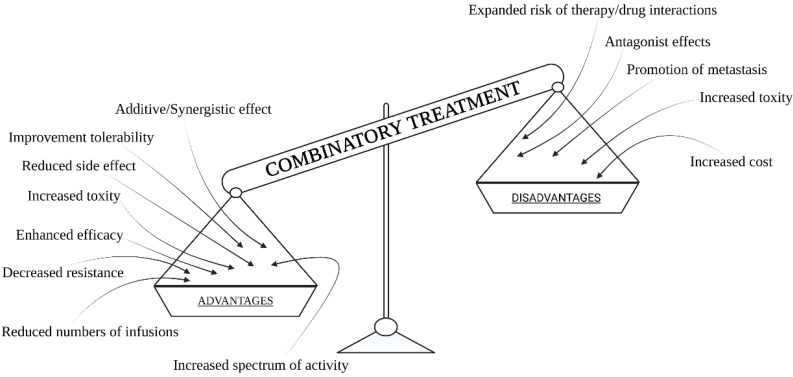
Pros and cons of combination treatment.

**Figure 2 pharmaceutics-13-01147-f002:**
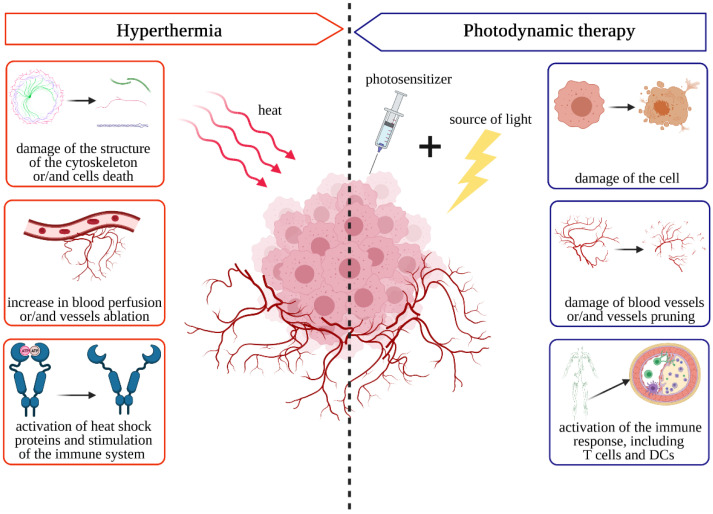
Effects of hyperthermia and PDT at the cellular, vascular, and immune levels.

**Table 1 pharmaceutics-13-01147-t001:** Completed, combined anticancer therapies; study started in the years 2015–2020 (ClinicalTrials.gov).

Cancer	Combination Therapy Scheme	Trial Design	Clinical Stage	Median OS * (Months)	Median PFS * (Months)	ClinicalTrial.gov ID	Ref.
Non-small cell lung cancer stage III	Chemoradiotherapy and immunotherapy	cis-/carboplatin + vinorelbine + etoposide/pemetrexed, radiation, and Nivolumab	II stage	38.8	12.7	NCT02434081 [23]	[24,25]
Unresectable non-small cell lung cancer	Chemotherapy and radiotherapy	Vinorelbine, Cisplatin, and radiation	II stage	35.6	11.5	NCT02709720 [26]	[27]
Small cell lung cancer	Chemotherapy and immunotherapy	Paclitaxel and Pembrolizumab	II stage	9.2	5.0	NCT02551432 [28]	[29,30]
Non-small Cell Lung Cancer	Chemotherapy and immunotherapy	Carboplatin/Paclitaxel and SB8 (A proposed bevacizumab biosimilar) Carboplatin/Paclitaxel and Bevacizumab	III stage	14.90 15.80	8.5 7.9	NCT02754882 [31]	[32]
Non-small Cell Lung Cancer	Chemotherapy and immunotherapy	Nivolumab and Ipilimumab EGFR Nivolumab plus Ipilimumab ALK Experimental: Nivolumab + Carboplatin + Pemetrexed with EGFR chemo-naive Nivolumab + Carboplatin + Pemetrexed ALK chemo-naive	II stage	22.3 7.6 7.75 5.9	1.3 0.7 4.65 2.8	NCT03256136 [33]	—
Small cell lung cancer	Immunology and radiotherapy	Durvalumab and Tremelimumab Durvalumab and Tremelimumab and hypofractionated radiotherapy/stereotactic body radiation therapy	II stage	2.8 5.7	2.1 3.3	NCT02701400 [34]	[35]
Non-small cell lung cancer	Target therapy and Immunotherapy	Ensartinib and Durvalumab	I/II stage	—	—	NCT02898116 [36]	—
Non-small cell lung cancer	Surgery and immunotherapy	Cryosurgery and NK Immunotherapy	I/II stage	—	—	NCT02843815 [37]	[38].
Lung neoplasms	Cryotherapy and target therapy	Cryotherapy and Icotinib	IV stage	—	—	NCT02744664 [39]	—
Non-small cell lung cancer	Chemotherapy and immunotherapy	Atezolizumab and Carboplatin and Paclitaxel (APC) Atezolizumab, Bevacizumab, Carboplatin, and Paclitaxel (ABCP) Bevacizumab, Carboplatin, and Paclitaxel (BCP)	III stage	In the ITT * population: ACP = 19.5 vs. ABCP = 19.8 vs. BCP = 14.9	In the ITT * population: ACP = - vs. ABCP = 8.4 vs. BCP = 6.8	NCT02366143 [40]	[41,42,43,44]
Small cell lung cancer	Chemotherapy and immunotherapy	Irinotecan and Dinutuximab	II/III stage	6.9	3.5	NCT03098030 [45]	[46]
Breast cancer	Chemotherapy and immunotherapy	Paclitaxel and Durvalumab	I/II stage	—	—	NCT02628132 [47]	—
Breast cancer	Chemotherapy and immunotherapy	Nab-paclitaxel and Durvalumab (MEDI4736) Epirubicin and Cyclophosphamide and Durvalumab (MEDI4736)	II stage	—	—	NCT02685059 [48]	[49]
Breast cancer	Chemotherapy and target therapy	Eribulin and PQR309	I/II stage	—	—	NCT02723877 [50]	[51]
Breast cancer	Chemotherapy and immunotherapy	Docetaxel and carboplatin and trastuzumab Epirubicin and cyclophosphamide followed by docetaxel and trastuzumab	II stage	—	—	NCT03140553 [52]	[53]
Breast cancer	Chemotherapy and immunotherapy	Cyclophosphamide and Pembrolizumab	II stage	—	—	NCT03139851 [54]	—
Breast cancer	Surgery and immunotherapy	Cryosurgery and NK immunotherapy	I/II stage	—	—	NCT02844335 [55]	—
Breast cancer	Chemotherapy and immunotherapy	Non-pegylated liposomal Doxorubicin and Trastuzumab	I stage	—	7.2	NCT02562378 [56]	[57,58]
Breast cancer	Chemotherapy and immunotherapy	Docetaxel and Pertuzumab and Trastuzumab	IV stage	—	23.0	NCT02445586 [59]	—
Breast cancer	Chemotherapy and immunotherapy	Eribulin and Durvalumab	I stage	—	—	NCT03430518 [60]	[61]
Breast cancer	Chemotherapy and immunotherapy	Docetaxel and Pertuzumab (Perjeta) and Trastuzumab (Herceptin)	III stage	—	18.7	NCT02402712 [62]	[63]
Breast cancer	Target therapy and immunotherapy	Ibrutinib and Durvalumab	I/II stage	4.2	1.7	NCT02403271 [64]	[65]
Breast cancer	Hormone therapy and targeted therapy	Letrozole and Palbociclib	IV stage	—	—	NCT02679755 [66]	—
Metastatic Breast Cancer	Chemotherapy and target therapy	Paclitaxel and S81694	I/II stage	—	—	NCT03411161 [67]	—
Breast cancer	Hormone therapy and targeted therapy	Letrozole and Nintedanib	I stage	—	—	NCT02619162 [68]	[69]
Breast cancer	Hormone therapy and targeted therapy	Letrozole and Ribociclib	II stage	—	—	NCT03248427 [70]	[71]
Breast Cancer	Chemotherapy and immunotherapy and hormone therapy	Epirubicin, Cyclophosphamide, Nivolumab, Triptorelin, Exemestane	II stage	—	—	NCT04659551 [72]	—
Breast Cancer Bone-dominant metastatic breast cancer	Immunotherapy and hormone therapy and radiopharmaceutical drug	Denosumab and Tamoxifen/Fulvestrant and Ra-223 dichloride	II stage	—	7.4 or 16 (bone-dominant metastases)	NCT02366130 [73]	[74]
Estrogen receptor positive breast cancer	Hormone therapy and targeted therapy	Tamoxifen and TAK-228	II stage	—	—	NCT02988986 [75]	[76,77]
Breast cancer	Chemotherapy and immunotherapy	Docetaxel and Pertuzumab and Trastuzumab	III stage	NA	14.5	NCT02896855 [78]	[79]
Prostate cancer	Hormone therapy and immunotherapy	Abiraterone and TRC105 Enzalutamide and TRC105	II stage	—	—	NCT03418324 [80]	
Prostate cancer	Immunotherapy and surgery	huJ591 and 89Zr-J591 and radical prostatectomy	I stage	—	—	NCT02693860 [81]	—
Castration resistant prostate cancer	Chemotherapy and cryoimmunotherapy and immunotherapy	Cyclophosphamide and Dendritic cell-based cryoimmunotherapy and Ipilimumab	I stage	—	5 (150 days)	NCT02423928 [82]	[83]
Prostate cancer	Hormone therapy and targeted therapy	Enzalutamide and LY3023414	II stage	—	7.5	NCT02407054 [84]	[85]
Castrate-resistant prostate cancer	Immunotherapy and radiopharmaceutical drug	Atezolizumab and Radium-223 Dichloride	I stage	16.3	3.0	NCT02814669 [86]	[87]
Prostate cancer	Hormone therapy and surgery	Apalutamide and Radical Prostatectomy	II stage	—	—	NCT03124433 [88]	—
Prostate cancer	Hormone therapy and target therapy	Prednisone and Apalutamide/Abiraterone Acetate and Niraparib	I stage	—	—	NCT02924766 [89]	[90,91]
Prostate carcinoma metastatic to the bone	Hormone therapy and radiopharmaceutical drug	Enzalutamide and Radium-223 Dichloride	II stage	—	—	NCT02507570 [92]	[93]
Prostate cancer	Hormone therapy and target therapy and radiotherapy	Leuprolide acetate/Goserelin acetate/Degarelix, PLX3397, Radiation Therapy	I stage	—	—	NCT02472275 [94]	—
Colon cancer	Chemotherapy and immunotherapy	TAS-102 (Trifluridine/tipiracil) and Panitumumab	I/II stage	—	5.8	NCT02613221 [95]	[96,97]
Colorectal cancer	Chemotherapy and target therapy	Hydroxychloroquine, Entinostat, Regorafenib	I/II stage	—	—	NCT03215264 [98]	—
Solid tumor Colorectal cancer	Immunotherapy and target therapy	Magrolimab and Cetuximab	I/II stage	9.5 7.6	3.6 1.9	NCT02953782 [99]	[100]
Metastatic colorectal cancer	Chemotherapy and target therapy	FOLFIRI and Aflibercept	II stage	12.6	7.4	NCT02970916 [101]	—
Metastatic colorectal cancer	Chemotherapy and target therapy	Pemetrexed and Erlotinib	II stage	7.3	2.5	NCT02723578 [102]	[103]
Metastatic colorectal cancer	Immunotherapy and target therapy	Spartalizumab and Regorafenib	I stage	—	—	NCT03081494 [104]	—
Metastatic colorectal cancer	Chemotherapy and immunotherapy	TAS-102 (Trifluridine/tipiracil) and Bevacizumab Capecitabine and Bevacizumab	II stage	18.0 16.2	9.2 7.8	NCT02743221 [105]	[106]
Microsatellite stable relapsed or refractory colorectal cancer	Immunotherapy and target therapy	Avelumab and Tomivosertib (eFT508)	II stage	—	—	NCT03258398 [107]	[108]
Colorectal neoplasms	Chemotherapy and immunotherapy	mFOLFOX6 and BI 695502	III stage	19.4	10.5	NCT02776683 [109]	—
Colorectal neoplasm	Chemotherapy and target therapy	mFOLFOX6 and Selinexor	I stage	—	—	NCT02384850 [110]	[111]
Refractory metastatic colorectal cancer	Chemotherapy and immunotherapy	TAS-102 (Trifluridine/tipiracil) and Nivolumab	II stage	—	2.2	NCT02860546 [112]	[113,114]
Colorectal cancer	Immunotherapy and target therapy	Durvalumab and Pexidartinib	I stage	—	—	NCT02777710 [115]	—
Colorectal cancer	Immunotherapy and target therapy	Atezolizumab and Cobimetinib	III stage	8.87	1.91	NCT02788279 [116]	[117]
Colorectal cancer	Immunotherapy and target therapy	Atezolizumab and Bevacizumab and Cobimetinib	I stage			NCT02876224 [118]	—
Colorectal neoplasms	Chemotherapy and target therapy	FOLFIRI and Cetuximab	III stage	—	11.4	NCT02484833 [119]	[120,121]
Metastatic colorectal cancer	Chemotherapy and immunotherapy	Irinotecan and AZD1775	I stage	—	—	NCT02906059 [122]	[123]
Metastatic colorectal cancer	Chemotherapy and target therapy	TAS-102 (Trifluridine/tipiracil) and Brontictuzumab	I stage	—	—	NCT03031691 [124]	—
Metastatic colorectal cancer	Chemotherapy and target therapy	FOLFIRI and OMP-131R10	I stage	—	—	NCT02482441 [125]	—
Colorectal cancer	Immunotherapy and target therapy	Pembrolizumab and AMG820	I/II stage	38.963	5.396	NCT02713529 [126]	[127]
Pancreatic adenocarcinoma (PDAC)	Local ablative therapy and immunotherapy	Irreversible electroporation (IRE) and allogeneic γδ T cells	I/II stage	14.5	11	NCT03180437 [128]	[129]
Pancreatic neoplasms	Local ablative therapy and immunotherapy	Irreversible electroporation (IRE) and NK cells	I/II stage	—	—	NCT02718859 [130]	—
Metastatic pancreatic cancer	Target therapy and Chemotherapy	RX-3117 (Fluorocyclopentenylcytosine) and Abraxane	I/II stage	—	—	NCT03189914 [131]	[132]
Advanced/ metastasized pancreatic cancer	Chemotherapy and target therapy	Gemcitabine, nab-paclitaxel, LED225 (Sonidegib)	I/II stage	6.0	—	NCT02358161 [133]	[134]
Pancreatic cancer	Chemotherapy and target therapy	Gemcitabine and BP31510 (Ubidecarenone, USP)	I stage	—	—	NCT02650804 [135]	[136]
Pancreatic cancer	Immunotherapy and target therapy	Pembrolizumab and Olaptesed pegol	I/II stage	—	1.87	NCT03168139 [137]	[138,139]
Pancreatic cancer	Immunotherapy and radiotherapy	Nivolumab and Cabiralizumab and Stereotactic Body Radiotherapy (SBRT)	II stage	—	—	NCT03599362 [140]	[141]
Pancreatic cancer	Chemotherapy and irreversible electroporation	Gemcitabine and Irreversible electroporation (IRE)	I stage	—	—	NCT02981719 [142]	—
Pancreatic cancer	Chemotherapy and immunotherapy	FOLFOX and Pegilodecakin	III stage	5.8	2.1	NCT02923921 [143]	[144]
Pancreatic cancer	Chemotherapy and immunotherapy	nab-Paclitaxel, Gemcitabine, ALT-803	I/II stage	—	—	NCT02559674 [145]	[146]
Metastatic pancreatic cancer	Immunotherapy and target therapy	Pembrolizumab and Acalabrutinib	II stage	—	1.4	NCT02362048 [147]	[148]
Metastatic pancreatic cancer	Immunotherapy and target therapy	Durvalumab and Galunisertib	I stage	5.72	1.87	NCT02734160 [149]	[150,151]
Pancreatic cancer	Chemotherapy and immunotherapy	nab-Paclitaxel, Gemcitabine, Selicrelumab	I stage	—	—	NCT02588443 [152]	—
Pancreatic cancer	Chemotherapy and radiotherapy	mFOLFIRINOX and Stereotactic Body Radiotherapy (SBRT)	II stage	—	—	NCT03891472 [153]	—
Pancreatic cancer	Chemotherapy and immunotherapy	Gemcitabine and M7824	I/II stage	—	—	NCT03451773 [154]	—
Pancreatic neoplasms	Chemotherapy and target therapy	nab-Paclitaxel, Gemcitabine, Napabucasin	III stage	—	—	NCT02993731 [155]	[156,157]

Table 1 completed, interventional clinical trials combined anticancer therapies; study started in the years 2015–2020 (ClinicalTrials.gov - last assessed on 26 April 2021). Information was retrieved from ClinicalTrials.gov, by searching for ‘condition or disease: ‘lung cancer’; ‘breast cancer’; ‘prostate cancer’; ‘colorectal cancer’; ‘pancreatic cancer’, status recruitment: ‘completed’, study type: ‘interventional studies (Clinical Trials)’. Study start: from 1 January 2015 to 31 December 2020. * OS—overall survival, * PFS—progression-free survival, NE—not estimable, * ITT—intention-to-treat.

**Table 2 pharmaceutics-13-01147-t002:** Temperature ranges for specific parts of the human body.

Temperature Range (°C)	Selected Area in the Human Body
36.32–37.76	rectal
35.76–37.52	tympanic
35.61–37.61	urine
35.73–37.41	oral
35.01–36.93	axillary

## Data Availability

Not applicable.

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
