# Peer review of "Photodynamic Therapy and Hyperthermia in Combination Treatment—Neglected Forces in the Fight against Cancer"

_pharmaceutics, 2021, doi:10.3390/pharmaceutics13081147_

Round 1
Reviewer 1 Report
This is a review article made by Bienia et al. to describe the application of photodynamic therapy (PDT) and hyperthermia in the combination cancer treatment. The authors did a comprehensive review describing the importance and the purpose of the combination of different cancer therapy. Moreover, the relevant clinical trials were listed clearly in the table. Based on this, the advantages of PDT and hyperthermia in the combination cancer therapy have been detailed illustrated in this article, which gives a whole picture of PDT and hyperthermia in cancer therapy. This review is quite useful for the study of alternative cancer therapy. I just have three minor opinions.
- The authors described a lot of the advantages of PDT and hyperthermia in the combination treatment, however, I can not find the description for the disadvantages of these two therapies. Please introduce them.
- In my opinion, the section describing why PDT or hyperthermia is a good option for combination treatment is the most valuable in this review. Because of too much information, I suggest the author could describe the information of PDT and hyperthermia separately in the different paragraphs under each reason to avoid misunderstanding.
- The abbreviation “PDT” should be given its full name when it was used firstly in the text.
Author Response
Dear Reviewer,
Thank you so much for Your helpful suggestions. We were happy to modify our manuscript with the following changes:
- The disadvantages of combinatory treatment are in section 1, especially in 1.1 subsection. The disadvantages of PDT are in lines 238-249, and hyperthermia is described in lines 300-314.
- Section 3 with information on why PDT and hyperthermia are good options for combinatory treatment was organized by us, to obtain better clarity.
- The abbreviation of PDT was introduced at the beginning of the introduction.
Thank You so much for Your help.
Reviewer 2 Report
Bienia et. al summarized the major biological effects of photodynamic therapy and hyperthermia and also touched the possible combination therapy based on these biological effects. It represents a interesting topic. Overall, it is well-organized. I recommended its publications after the following issues are addressed.
1.The authors should discuss "how to implement combination therapy".
2.In figure 2, the authors discussed the disadvantages of combination therapy. But they did not provided the possible strategies to solve these issues. Especially, the toxicity should be a major issue.
3.The authors should add another section to discuss the effects of photodynamic therapy and hyperthermia on nanomedicine (e.g., Science, 2003, 300, 595-596; J. Am. Chem. Soc. 2021, 143, 2, 538–559). It might improve the functionality of nanomedicine. Particularly, photodynamic effect and hyperthermia could increase the responsiveness of smart nanomedicine, which is now considered as a significant problem for nanomedicine (e.g. Angewandte Chemie International Edition, 2020, 59(32): 13526-13530).
4.Another important biological effect is increasing receptor expression for active targeting including nanomedicine (Trends Mol. Med. 2002, 8 (12), 563−71; J. Controlled Release 2018, 275, 142−61.).
5. The authors should discuss the possible side effects of photodynamic therapy and hyperthermia, such as promoting the metastasis because of permeability increase (Cancer Lett. 2013, 335 (2), 259−69.).
6.PDT and hyperthermia also can be combined. Particularly, hyperthermia could solve the issue of hypoxia for PDT (e.g.,Advanced Functional Materials, 2017, 27(33): 1702108).
7.In Graphical Abstract, 'chemiotherapy' should be 'chemotherapy'.
Author Response
Dear Reviewer,
Thank You so much for Your very helpful comments and suggestions. We propose the following changes to the manuscript:
- The paragraph regarding therapy implementation can be found in paragraph 4.2.
- In our opinion solving all disadvantages of combinatory treatment can be made by individualized treatment plans created on knowledge about mechanisms of actions of particular treatment and interactions between them. That is why, in fig 2 the symbol of the scale was used - to explain that finding a balance is crucial for combinatory treatment effectiveness. The ways to obtain this balance were described in section 4, especially 4.1.
- We added another section about nanomedicine, to explain its importance in current research and show clinical trends- it could be found in section 3. We believe that nanoparticles can solve a lot of problems and can bring new solutions to cancer treatment e.g. theranostic nanomaterials have a great potential to perform diagnosis and treatment at the same time. Another interesting observation is how to presence of nanoparticles can change the expression of a specific receptor.
- The effect of increased receptor expression was described in lines 606-618
- We discussed the influence of combinatory treatment on metastasis in lines 84-88. It should be clarified that non-optimal combinatory treatment can be harmful to the patients. One of the examples should be tumor reoxygenation, which can occur after some PDT and hyperthermia protocols, which we named “therapeutic window”. This period of time with increased pO2 level in the tumor can be used for radiotherapy to take an advantage of the oxygen enchantment effect (well-oxygenated tissue is responding to low LED radiation three times better than hypoxic ones). However, this “therapeutic window” can be a “boost” to tumor tissue and accelerate tumor growth. At the same time, metastatic cells can reach blood vessels due to the possible enchantment of endothelial cell permeability (EPR effect due to PDT hyperthermia).
- Another section about the combination of PDT and hyperthermia together introduced at section 6.
- The figures and graphical abstract were checked for misspelling words.
Thank you for your help.
Reviewer 3 Report
The scientific quality of the paper is very poor if any. Here I mention just few examples to support my opinion.
Title is misleading. The paper presents a kind of review of PDT and hyperthermia and not a review of their combination therapy.
The text is wordy, redundant. It provides a very superficial information about modalities discussed.
Authors have a very poor understanding about photodynamic therapy. Meaning of several sentences, such as:
…”it is activated by an appropriate wavelength of light that interacts with oxygen, thus creating a photodynamic reaction..”
…”They may be 352 slightly photosensitizing even without the use of light”
is just scientific nuisance.
Spelling of terminology are careless:
weather instead of water
days instead of dyes
PDT therapy instead of PDT (photodynamic therapy)
Several statements of the paper are not supported by scientific results, e.g.,
“…1 of 3 cancer patients is uncurable”.
This simplification is an unallowable and dangerous.
Author Response
Dear Reviewer,
- We are proposing a new title for our manuscript:
Photodynamic therapy and hyperthermia in combination treatment
– neglected forces in the fight against cancer - We designed this manuscript for students and researchers who are working in different fields and are interested in familiarizing themselves with combinatory therapies against cancer. In our opinion, one of the problems, why PDT and hyperthermia are not so popular in the clinical practice, is the fact, that deep knowledge in different fields is needed to plan such a combinatory therapy e.g. about laser and light dosimetry, radiotherapy, and dose planning, the kinetics of drugs and pro-drugs accumulation, temperature monitoring deep in the tissue, mechanisms of actions… That is why, we are discussing all important factors from the basic level - to help introduce all those needs to people who are interested in combinatory treatment. We hope that after reading our manuscript, it will be clear why combinatory treatment should be interdisciplinary teamwork.
- Thank You for pointing out our mistakes in grammar and word spelling. We corrected unclear and misleading sentences to provide more precise information.
- We changed the sentence “1 of 3 cancer patients is uncurable” to “for 1 of 3 patients cancer is leading to death” to remain clear.
Thank You for Your comments.
Round 2
Reviewer 3 Report
In spite of considerable changes made in the text, scientific significance and novelty of the paper has not been improved.